# Alignment of Diffusion Model and Flow Matching for Text-to-Image Generation

## Abstract

Diffusion models and flow matching have demonstrated remarkable success in text-to-image generation. While many existing alignment methods primarily focus on fine-tuning pre-trained generative models to maximize a given reward function, these approaches require extensive computational resources and may not generalize well across different objectives. In this work, we propose a novel alignment framework by leveraging the underlying nature of the alignment problem—sampling from reward-weighted distributions—and show that it applies to both diffusion models (via score guidance) and flow matching models (via velocity guidance). We show that the score function (velocity field) required for the reward-weighted distribution can be decomposed into the pre-trained score (velocity field) plus a conditional expectation of the reward. For the alignment on the diffusion model, we identify a fundamental challenge: the adversarial nature of the guidance term can introduce undesirable artifacts in the generated images. Therefore, we propose a finetuning-free framework that trains a guidance network to estimate the conditional expectation of the reward. We achieve comparable performance to finetuning-based models with one-step generation with at least a 60% reduction in computational cost. For the alignment on flow matching, we propose a training-free framework that improves the generation quality without additional computational cost.

## 1 Introduction

Diffusion models and flow matching have achieved impressive performance in text-to-image generation, as demonstrated by state-of-the-art models such as Imagen (Saharia et al., 2022), DALL-E 3 (Betker et al., 2023), Stable Diffusion (Rombach et al., 2021; Podell et al., 2024), and rectified flow model (Esser et al., 2024). These models have been proven capable of generating high-quality, creative images even from novel and complex text prompts. In practice, one may want to further align these pre-trained generators with a desired preference or reward, such as human preference, aesthetic quality, object attributes, spatial relations, or other user-specified criteria (Kirstain et al., 2023; Wu et al., 2023; Xu et al., 2023; Ghosh et al., 2023). In this work, we study the alignment of pre-trained text-to-image generators, where the goal is to steer an existing generator toward a specified reward function without training the generator from scratch.

Inspired by Reinforcement Learning from Human Feedback (RLHF) (Ouyang et al., 2022), many alignment approaches leverage preference pairs to fine-tune models for generating samples that align with task-specific objectives. RLHF-type methods typically learn a reward function and use the policy gradients to update the model (Lee et al., 2023; Fan et al., 2023; Black et al., 2024; Clark et al., 2024; Chakraborty et al., 2024; Jaques et al., 2016; Liu et al., 2025; Jaques et al., 2020). On the other hand, Direct Preference Optimization (DPO)-type methods directly optimize the model to adhere to human preferences, without requiring explicit reward modeling or reinforcement learning (Rafailov et al., 2024; Wallace et al., 2023; Yang et al., 2023; Liang et al., 2025; Yang et al., 2024).

Despite their effectiveness, these approaches require modifying model parameters through fine-tuning, which comes with several potential limitations. For example, fine-tuning for new reward functions is computationally expensive and often requires carefully designed training strategies; otherwise, optimizing on a limited set of

input prompts can limit generalization to unseen prompts. More importantly, existing fine-tuning approaches do not fully exploit the structure of the alignment problem. Instead, they typically apply Low-Rank Adaptation (LoRA) to optimize model weights for a specific reward function (Hu et al., 2022), which may not be the most efficient strategy.

In contrast, plug-and-play alignment methods integrate new objectives without modifying the underlying model parameters, significantly reducing computational costs while adapting flexibly to different reward functions. This idea has been widely used in classifier or classifier-free guidance (Dhariwal & Nichol, 2021; Ho, 2022), inverse-problem Chung et al. (2023), and energy-guided diffusion sampling (Graikos et al., 2022; Song et al., 2023a; Bansal et al., 2023; Yu et al., 2023; Shen et al., 2024; Ye et al., 2024). These methods suggest that pre-trained diffusion models can be steered at sampling time, but they do not directly address reward alignment for text-to-image generation.

In this paper, we cast alignment for both diffusion models and flow matching models as a unified sampling problem from reward-weighted distributions. Specifically, given a pre-trained reference model and a reward function, the reward-weighted distribution reweights the reference distribution in proportion to the exponentiated reward, so that higher-reward images are assigned larger probability. Thus, instead of finetuning the backbone generator to optimize the reward, we directly modify the sampling dynamics toward this reward-weighted distribution. Under this formulation, we show that the key object needed for sampling, the new score function for diffusion or the new velocity field for flow matching, can be decomposed as the corresponding pre-trained quantity plus an additional reward-driven guidance term.

For diffusion models, this decomposition gives a score-guidance formula: the aligned score equals the pre-trained score plus the gradient of the log conditional expectation of the exponentiated reward. However, we observe that this guidance term admits an adversarial nature flaw, as it is a gradient with respect to the high-dimensional input space and may introduce undesirable artifacts when directly applied during generation. To address this issue, we propose a finetuning-free [1] alignment method that trains a lightweight guidance network to estimate the required conditional expectation, together with a regularization strategy that stabilizes the guidance landscape. We evaluate the effectiveness of the proposed method on four widely used criteria for text-to-image generation, and the proposed method achieves comparable performance to finetuning-based models in one-step generation while reducing computational cost by at least 60%.

For flow matching models, we derive the exact form of velocity guidance. In this case, the aligned velocity field equals the pre-trained velocity field plus a reward-driven conditional expectation of velocity fields. Unlike the diffusion guidance term, the correction here is a conditional expectation rather than a gradient with respect to the high-dimensional input space. Based on this structure, we propose a training-free estimator that directly computes the guidance term without additional model fine-tuning. The proposed method improves the generation quality without additional training overhead.

Our contributions are summarized as follows.

- We formulate alignment of diffusion models and flow matching models as sampling from reward-weighted distributions, yielding a unified plug-and-play view of reward alignment.
- For diffusion models, we derive a score-guidance formula and propose a finetuning-free method that trains a lightweight guidance network with a regularized guidance landscape.
- For flow matching models, we derive the exact reward-driven velocity guidance and propose a training-free estimator that improves generation without additional model finetuning.
  used preference and quality metrics. In one-step diffusion generation, it achieves performance comparable to finetuning-based baselines while reducing computational cost by at least 60%.

---

[1] Finetuning-free refers specifically to the pretrained backbone model. Our approach does not modify the backbone, but instead trains a separate guidance network.

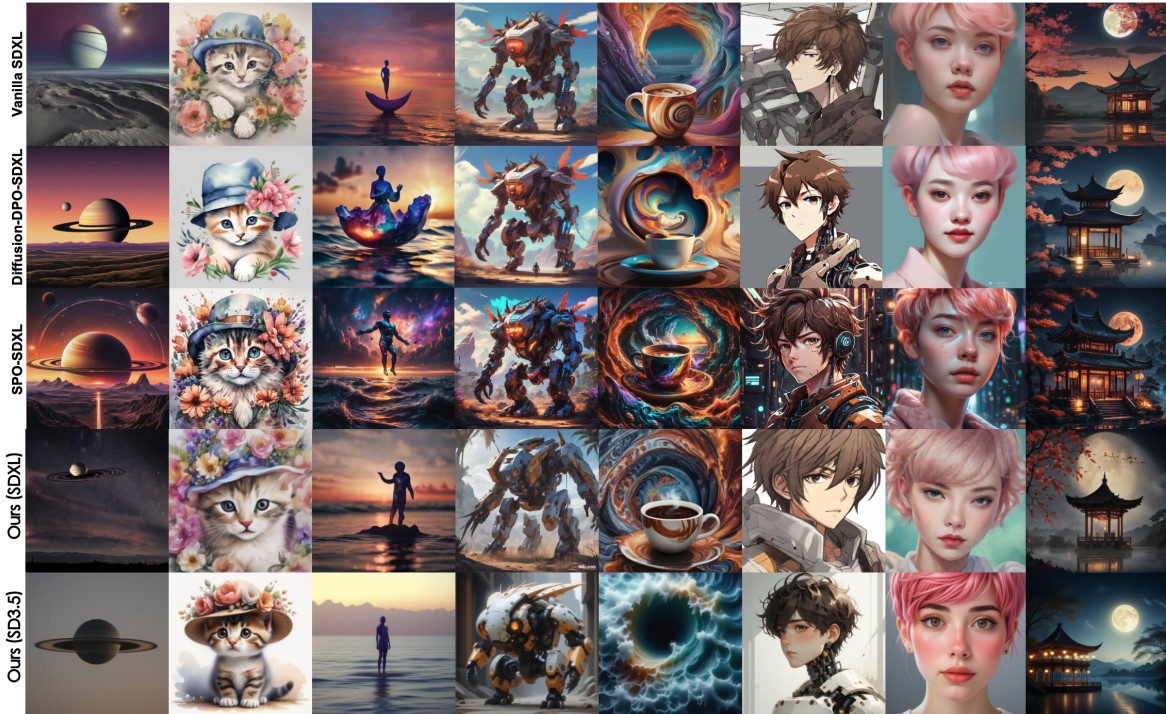

Figure 1: Qualitative comparison with Vanilla SDXL, Diffusion-DPO, and SPO. Our method achieves better aesthetic quality and stronger alignment with the text prompt. Prompts are provided in the Appendix B.4.

## 2 Preliminaries

In this section, we begin with a brief overview of diffusion models and flow matching in Section 2.1 and Section 2.2. We then review existing techniques for aligning pre-trained models with human preferences, and decompose the alignment procedure into two key components: reward learning for modeling human preferences in Section 2.3 and the alignment methods in Section 2.4.

### 2.1 Diffusion Models

Diffusion generative models are characterized by their forward and backward processes (Ho et al., 2020; Song et al., 2021). The forward process gradually injects Gaussian noise into samples $\mathbf{x}_0$ from the data distribution $p_0$, following the stochastic differential equation:

$$\mathrm{d}\mathbf{x}_t = \mathbf{f}(\mathbf{x}_t, t)\mathrm{d}t + g(t)\mathrm{d}\mathbf{w}, \ t \in [0, T], \tag{1}$$

where $\mathbf{w}$ is the standard Brownian motion, $\mathbf{f}(\cdot, t) : \mathbb{R}^d \to \mathbb{R}^d$ is a drift coefficient, and $g(\cdot) : \mathbb{R} \to \mathbb{R}$ is a diffusion coefficient. We use $p_t(\mathbf{x})$ to denote the marginal distribution of $\mathbf{x}_t$ at time $t$. And we can use the time reversal of Eq. (1) for generation, which admits the following form (Anderson, 1982):

$$\mathrm{d}\mathbf{x}_t = \left[\mathbf{f}(\mathbf{x}_t, t) - g(t)^2 \nabla_{\mathbf{x}} \log p_t(\mathbf{x})\right] \mathrm{d}t + g(t)\mathrm{d}\overline{\mathbf{w}}, \tag{2}$$

where $\overline{\mathbf{w}}$ is a standard Brownian motion when time flows backwards from $T$ to $0$, and $\mathrm{d}t$ is an infinitesimal negative time step. The score function of each marginal distribution $\nabla_{\mathbf{x}} \log p_t(\mathbf{x})$ needs to be estimated by the following score matching objective:

$$\min_{\boldsymbol{\theta}} \ \mathbb{E}_t \left\{\lambda(t)\mathbb{E}_{p_t(\mathbf{x}_t)}\left[\|\mathbf{s}_{\boldsymbol{\theta}}(\mathbf{x}_t, t) - \nabla_{\mathbf{x}_t} \log p_t(\mathbf{x}_t)\|_2^2\right]\right\}, \tag{3}$$

where $\lambda(t) : [0, T] \to \mathbb{R}_{>0}$ is a positive weighting function, $t$ is uniformly sampled over $[0, T]$. The latent diffusion model (Rombach et al., 2021; Podell et al., 2024) further extends diffusion models to text-to-image

generation. They use an image encoder that maps $\mathbf{x}$ into a latent representation and use a text encoder that maps the prompts $y$ into an embedding as the condition.

## 2.2 Flow Matching

Flow matching models learn a time-dependent velocity field that transports a simple base distribution to the data distribution (Lipman et al., 2023) via the probability flow ODE:

$$\frac{\mathrm{d}\mathbf{x}_t}{\mathrm{d}t} = \mathbf{v}_{\boldsymbol{\phi}}(\mathbf{x}_t, y, t), \quad t \in [0, 1],$$

where $\mathbf{v}_{\boldsymbol{\phi}} : \mathbb{R}^d \to \mathbb{R}^d$ is a learnable velocity field. Unlike diffusion models, we denote $\mathbf{x}_0$ as a sample from a base distribution (e.g., standard Gaussian) and $\mathbf{x}_1$ as a sample from the data distribution.

The flow matching objective minimizes the discrepancy between the model vector field and the oracle velocity field along the trajectory:

$$\mathcal{L}(\theta) = \mathbb{E}_{\mathbf{x}_t \sim p_t(\mathbf{x}_t | \mathbf{x}_0, \mathbf{x}_1), t \sim \mathcal{U}[0,1]} \left[ \left\| \mathbf{v}_{\boldsymbol{\phi}}(\mathbf{x}_t, t) - \mathbf{v}(\mathbf{x}_t, y, t) \right\|_2^2 \right], \tag{4}$$

where $\mathbf{x}_t$ is a linear interpolation between $\mathbf{x}_0$ and $\mathbf{x}_1$, and $\mathbf{v}(\mathbf{x}_t, y, t)$ is the oracle velocity field.

## 2.3 Reward Learning

The Bradley-Terry (BT) model (Bradley & Terry, 1952), and the more general Plackett-Luce ranking models (Plackett, 1975; Luce, 1959), are commonly used to model preferences. Given a prompt $y$ and a pair of responses $\mathbf{x}_w \succ \mathbf{x}_l \mid y$, where $\mathbf{x}_w$ denotes the winning response and $\mathbf{x}_l$ denotes the losing response under the preference of humans. The BT model depicts the preference distribution as

$$p(\mathbf{x}_w \succ \mathbf{x}_l \mid y) = \frac{\exp(r(\mathbf{x}_w, y))}{\exp(r(\mathbf{x}_w, y)) + \exp(r(\mathbf{x}_l, y))},$$

where $r(\mathbf{x}, y)$ denotes the reward model and can be learned by the following maximum likelihood objective,

$$\min_{\boldsymbol{\phi}} - \mathbb{E}_{(\mathbf{x}_w, \mathbf{x}_l, y) \sim \mathcal{D}} \left[ \log \sigma \left( r(\mathbf{x}_w, y) - r(\mathbf{x}_l, y) \right) \right], \tag{5}$$

where $\mathcal{D} = \{\mathbf{x}_w^{(i)}, \mathbf{x}_l^{(i)}, y^{(i)}\}_{i=1}^N$ is the offline preference dataset and $\sigma$ denotes the logistic function.

## 2.4 Alignment

Building on the success of alignment techniques for finetuning large pre-trained models, many studies have explored aligning diffusion models and flow matching with human preferences. We review these approaches in the following.

**Reinforcement Learning from Human Feedback.** This type of works (Lee et al., 2023; Xu et al., 2023; Fan et al., 2023; Black et al., 2024; Clark et al., 2024) finetune the pre-trained model $\pi_{\mathrm{ref}}$ by policy gradient objectives (Jaques et al., 2016; 2020). In particular, the fine-tuned model $\pi_\theta$ is obtained by solving the following optimization problem:

$$\max_{\pi_\theta} \mathbb{E}_{y \sim \mathcal{D}_{\mathrm{prompt}}, \mathbf{x} \sim \pi_\theta(\mathbf{x}|y)} \left[ r(\mathbf{x}, y) \right] - \beta \mathbb{D}_{\mathrm{KL}} \left[ \pi_\theta(\mathbf{x} \mid y) \| \pi_{\mathrm{ref}}(\mathbf{x} \mid y) \right], \tag{6}$$

where $\mathcal{D}_{\mathrm{prompt}}$ denotes the prompt dataset, $\mathbb{D}_{\mathrm{KL}}$ represent the Kullback-Leibler (KL) divergence between the fine-tuned model and the reference model, and $\beta \geq 0$ controls the strength of the KL regularization. This type of method requires a pre-trained reward function for policy optimization (Schulman et al., 2017).

**Direct Preference Optimization.** Rafailov et al. (2024) propose not to explicitly learn the reward function. They start with the analytic solution of Eq. (6) as the energy-guided form,

$$\pi_{\mathrm{r}}(\mathbf{x} \mid y) = \frac{1}{Z(y)} \pi_{\mathrm{ref}}(\mathbf{x} \mid y) \exp\left(\frac{1}{\beta} r(\mathbf{x}, y)\right), \tag{7}$$

where $Z(y) = \int \pi_{\mathrm{ref}}(\mathbf{x} \mid y) \exp\left(\frac{1}{\beta} r(\mathbf{x}, y)\right) \mathrm{d}\mathbf{x}$ is the partition function. Therefore, they can reparameterize the reward function $r(\mathbf{x}, y)$ as

$$r(\mathbf{x}, y) = \beta \log \frac{\pi_{\mathrm{r}}(\mathbf{x} \mid y)}{\pi_{\mathrm{ref}}(\mathbf{x} \mid y)} + \beta \log Z(y). \tag{8}$$

Plugging Eq. (8) into Eq. (5) yields the objective of DPO-type methods:

$$\min -\mathbb{E}_{(\mathbf{x}_w, \mathbf{x}_l, y) \sim \mathcal{D}} \left[ \log \sigma\left(\beta \log \frac{\pi_\theta(\mathbf{x}_w \mid y)}{\pi_{\mathrm{ref}}(\mathbf{x}_w \mid y)} - \beta \log \frac{\pi_\theta(\mathbf{x}_l \mid y)}{\pi_{\mathrm{ref}}(\mathbf{x}_l \mid y)}\right) \right]. \tag{9}$$

## 3 Proposed Framework for Diffusion Model

In this section, we introduce a finetuning-free framework to directly sample from the reward-guided distribution for diffusion models. We begin by introducing the methodology formulation in Section 3.1. We then provide an in-depth analysis of several vanilla methods for calculating the guidance in Section 3.2. We highlight that these vanilla guidance methods exhibit adversarial guidance, which generates undesirable artifacts and worsens performance, particularly in text-to-image generation. Then, we present an enhanced method in Section 3.3 that alleviates the problem. All proofs are deferred to Appendix A.

### 3.1 Methodology Formulation

Inspired by previous works from transfer learning (Ouyang et al., 2024), we consider preference learning in terms of transferring a pre-trained diffusion model to adapt to the given preference data. To this end, we propose a finetuning-free alignment method for diffusion models. Instead of using RLHF-type (like Eq. (6)) or DPO-type (like Eq. (9)) alignments, we propose to directly sample from the reward-weighted distribution $\pi_{\mathrm{r}}(\mathbf{x}|y)$ in Eq. (7) leveraging the relationships between score functions in the following Theorem.

**Theorem 3.1.** *Let the conditional distribution of reference diffusion model $\pi_{ref}(\mathbf{x}|y)$ be denoted as distribution $p$ and the reward-weighted distribution $\pi_r(\mathbf{x}|y)$ defined in Eq. (7) as distribution $q$. Under some mild assumption of the forward noising process detailed in Appendix A, let $\phi^*$ be the optimal solution for the conditional diffusion model trained on target domain $q(\mathbf{x}_0, y)$, i.e.,*

$$\phi^* = \arg\min_{\phi} \mathbb{E}_t \left\{ \lambda(t) \mathbb{E}_{q_t(\mathbf{x}_t, y)} \left[ \left\| \mathbf{s}_\phi(\mathbf{x}_t, y, t) - \nabla_{\mathbf{x}_t} \log q_t(\mathbf{x}_t|y) \right\|_2^2 \right] \right\},$$

*then*

$$\mathbf{s}_{\phi^*}(\mathbf{x}_t, y, t) = \underbrace{\nabla_{\mathbf{x}_t} \log p_t(\mathbf{x}_t|y)}_{\substack{\text{pre-trained conditional model} \\ \text{on source}}} + \underbrace{\nabla_{\mathbf{x}_t} \log \mathbb{E}_{p(\mathbf{x}_0|\mathbf{x}_t, y)} \left[ \exp(\frac{1}{\beta} r(\mathbf{x}_0, y)) \right]}_{\text{conditional guidance}}. \tag{10}$$

Based on Eq. (10), we can calculate the additional guidance term rather than finetuning the text-to-image generative model. In general, the guidance term in Eq. (10) is not straightforward to compute as we need to sample from $p(\mathbf{x}_0|\mathbf{x}_t, y)$ for each $\mathbf{x}_t$ in the generation process. In the following, we first discuss some existing ways to calculate the guidance term.

### 3.2 Vanilla Method to Compute the Guidance Term

**Method 1: Direct backpropagation through diffusion process.** The first method directly backpropagates through diffusion process to calculate $\nabla_{\mathbf{x}_t} \log \mathbb{E}_{p(\mathbf{x}_0|\mathbf{x}_t, y)}[\exp(r(\mathbf{x}_0, y)/\beta)]$ for fine-tuning the diffusion

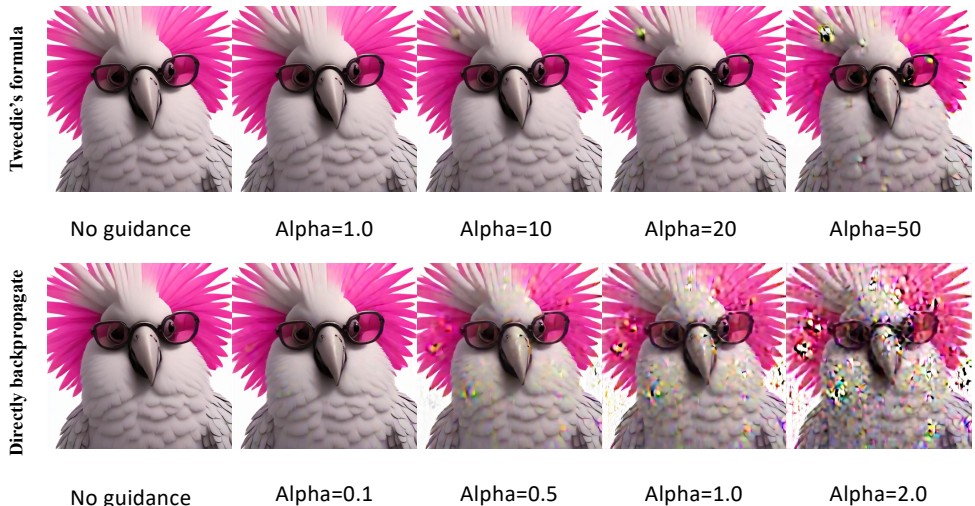

Figure 2: Illustration of the Adversarial Nature of Guidance. When the strength of the guidance is too small, there is little difference between the generated images with or without guidance. However, as the magnitude of the guidance increases (from left to right), undesirable artifacts become more pronounced. The prompt is "A 3D Rendering of a cockatoo wearing sunglasses. The sunglasses have a deep black frame with bright pink lenses. Fashion photography, volumetric lighting, CG rendering."

model. In Song et al. (2023a), the authors propose an unbiased Monte Carlo estimation:

$$\nabla_{\mathbf{x}_t} \log \mathbb{E}_{p(\mathbf{x}_0|\mathbf{x}_t, y)}\Big[ \exp\Big( \frac{1}{\beta} r(\mathbf{x}_0, y)\Big)\Big] \approx \nabla_{\mathbf{x}_t} \log \frac{1}{n} \sum_{i=1}^{n} \exp\Big( \frac{1}{\beta} r(\mathbf{x}_0^i, y)\Big),$$

where $\mathbf{x}_0^i$ denotes the $i$-th sample drawn from $p(\mathbf{x}_0|\mathbf{x}_t, y)$. However, this Monte Carlo estimation significantly increases memory costs, especially in text-to-image generation. Inspired by recent studies (Clark et al., 2024), we can borrow the same techniques, e.g., accumulated gradients along the diffusion process using techniques such as low-rank adaptation (LoRA) (Hu et al., 2022) and truncation or gradient checkpointing (Prabhudesai et al., 2023; Clark et al., 2024), to alleviate the memory cost of backpropagating through the diffusion process for calculating the guidance term. We can further reduce the memory cost by using the few-step diffusion model as the reference model. Despite these techniques, the memory requirements remain higher compared to our proposed approach in Section 3.3.

**Method 2: Approximate and apply Tweedie's formula.** The second method first approximates the guidance term by (Chung et al., 2023):

$$\nabla_{\mathbf{x}_t} \log \mathbb{E}_{p(\mathbf{x}_0|\mathbf{x}_t, y)}\big[ \exp\big( \frac{1}{\beta} r(\mathbf{x}_0, y)\big)\big] \approx \frac{1}{\beta} \nabla_{\mathbf{x}_t} r\big(\mathbb{E}_{p(\mathbf{x}_0|\mathbf{x}_t, y)}[\mathbf{x}_0], y\big). \tag{11}$$

Then, Tweedie's formula is further applied as (Bansal et al., 2023; Chung et al., 2023; Yu et al., 2023):

$$\mathbb{E}\left[\boldsymbol{x}_0 \mid \boldsymbol{x}_t, y\right] = \boldsymbol{x}_t + \sigma_t^2 \nabla_{\boldsymbol{x}_t} \log p_t\left(\boldsymbol{x}_t|y\right).$$

However, as noted in Lu et al. (2023); Song et al. (2023a), the approximation used in Eq. (11) is biased, leading to an incorrect calculation of the guidance term.

We empirically evaluate the effectiveness of these two vanilla methods for aligning text-to-image generation tasks. Figure 2 illustrates their performance under the guidance of PickScore (Kirstain et al., 2023), a reward function that evaluates whether the generated images align with human aesthetic and semantic

Table 1: Comparison of finetuning-free alignment algorithms on diffusion models. Our method uniquely provides theoretical guarantees for the correct form for guidance with a step size guarantee.

| Method | Classifier Guidance | Direct backpropagate (M1) | Tweedie's formula (M2) | Ours |
|---|---|---|---|---|
| Formulation | $\frac{1}{\beta}\nabla_{\mathbf{x}_t} r(\mathbf{x}_t, y)$ | $\nabla_{\mathbf{x}_t}\log\frac{1}{n}\sum_{i=1}^{n}\exp\left(\frac{1}{\beta}r(\mathbf{x}_0^i, y)\right)$ | $\frac{1}{\beta}\nabla_{\mathbf{x}_t} r(\mathbb{E}_{p(\mathbf{x}_0|\mathbf{x}_t, y)}[\mathbf{x}_0], y)$ | $\nabla_{\mathbf{x}_t}\log h_{\boldsymbol{\psi}^*}(\mathbf{x}_t, y, t)$ |
| Unbiased | ✗ | ✓ | ✗ | ✓ |
| Step size guarantee | ✗ | ✗ | ✗ | ✓ |

preferences. The x-axis represents the strength of the guidance term, denoted by $\alpha^2$. Our experiments reveal a previously overlooked issue in tuning this hyperparameter $\alpha$, which may contribute to suboptimal alignment performance. As shown in Figure 2, insufficient values of $\alpha$ produce results nearly indistinguishable from unguided generation, while excessive values introduce substantial artifacts that degrade image quality.

We attribute this phenomenon to the adversarial nature of the guidance mechanism, as observed in prior work (Shen et al., 2024). In Eq. (10), the guidance term is directly added to the estimated score. If the landscape is not smooth enough or does not behave well[3], the adversarial nature of the guidance can lead to undesirable artifacts in the generated images. To address these limitations, our proposed framework in Section 3.3 provides theoretical guarantees for generating properly aligned distributions with a fixed strength parameter $\alpha = 1$. Furthermore, we develop an additional regularization technique for training the guidance network that mitigates these instability issues.

### 3.3 Proposed Finetuning-free Guidance for Diffison Models

We first utilize the following trick to calculate the conditional expectation, similar to previous works (Ouyang et al., 2024; Lu et al., 2023).

**Lemma 3.2.** *For a neural network $h_{\boldsymbol{\psi}}(\mathbf{x}_t, y, t)$ parameterized by $\boldsymbol{\psi}$, define the objective*

$$\mathcal{L}_{guidance}(\boldsymbol{\psi}) := \mathbb{E}_{p(\mathbf{x}_0, \mathbf{x}_t, y)}\left[\left\|h_{\boldsymbol{\psi}}(\mathbf{x}_t, y, t) - \exp(\frac{1}{\beta}r(\mathbf{x}_0, y))\right\|_2^2\right], \tag{12}$$

*then its minimizer $\boldsymbol{\psi}^* = \underset{\boldsymbol{\psi}}{\arg\min}\ \mathcal{L}_{guidance}(\boldsymbol{\psi})$ satisfies:*

$$h_{\boldsymbol{\psi}^*}(\mathbf{x}_t, y, t) = \mathbb{E}_{p(\mathbf{x}_0|\mathbf{x}_t, y)}\left[\exp(\frac{1}{\beta}r(\mathbf{x}_0, y))\right].$$

By Lemma 3.2, we can instead estimate the value $\mathbb{E}_{p(\mathbf{x}_0|\mathbf{x}_t, y)}[\exp(r(\mathbf{x}_0, y)/\beta)]$ using the guidance network $h_{\boldsymbol{\psi}^*}$ obtained by minimizing the objective function $\mathcal{L}_{\text{guidance}}(\boldsymbol{\psi})$, which can be approximated by easy sampling from the joint distribution $p(\mathbf{x}_0, \mathbf{x}_t, y)$. Then, the estimated score function for the aligned diffusion model can be calculated as follows:

$$\mathbf{s}_{\boldsymbol{\phi}^*}(\mathbf{x}_t, y, t) = \underbrace{\nabla_{\mathbf{x}_t}\log p(\mathbf{x}_t|y)}_{\substack{\text{pre-trained model} \\ \text{on source}}} + \underbrace{\nabla_{\mathbf{x}_t}\log h_{\boldsymbol{\psi}^*}(\mathbf{x}_t, y, t)}_{\text{guidance network}}. \tag{13}$$

To alleviate the adversarial nature of the guidance, we also introduce a consistency regularization term,

$$\mathcal{L}_{\text{consistence}} := \mathbb{E}_{q(\mathbf{x}_0, y)}\mathbb{E}_{q(\mathbf{x}_t|\mathbf{x}_0)}\left[\left\|\nabla_{\mathbf{x}_t}\log p(\mathbf{x}_t|y) + \nabla_{\mathbf{x}_t}\log h_{\boldsymbol{\psi}}(\mathbf{x}_t, y, t) - \nabla_{\mathbf{x}_t}\log q(\mathbf{x}_t|\mathbf{x}_0, y)\right\|_2^2\right], \tag{14}$$

to learn the guidance network $h_{\boldsymbol{\psi}^*}$ better, since the gradient of $\mathcal{L}_{\text{consistence}}$ with respect to $\mathbf{x}_t$ match the score in preferred data. The key point of this regularization is that we cannot easily change the landscape of a

---

[2]Although there is no $\alpha$ in Eq. (10), many guidance methods (Lu et al., 2023; Song et al., 2023a) add this hyperparameter in practice to balance the strength of the guidance term with the score.

[3]We use landscape to describe the change of reward given the change of images.

given predetermined reward function, but we can regularize the landscape of the learned guidance network to ensure the generation of high-quality images.

Combining the consistency regularization terms together with the original guidance loss in Eq. (12), the final learning objective for the guidance network can be described as follows:

$$\boldsymbol{\psi}^* = \arg\min_{\boldsymbol{\psi}} \{\mathcal{L}_{\text{guidance}} + \eta \, \mathcal{L}_{\text{consistence}}\}, \tag{15}$$

where $\eta \geq 0$ are hyperparameters that control the strength of additional regularization, which also enhances the flexibility of our solution scheme.

### 3.4 Further Improvement to One-step Generation

The training objectives in Eq. (12) and Eq. (14) are agnostic to the reference model, indicating that we can use any pre-trained diffusion model with any reward function, whether differentiable or not. Motivated by the computational efficiency of one-step generative models in practical applications (Song et al., 2023b; Geng et al., 2025), we further present a straightforward approach for applying our proposed finetuning-free guidance to one-step text-to-image models.

Specifically, instead of sampling $t$ uniformly from $[0, T]$, we can simply set $t = T$. This small modification offers several advantages. First, while one-step diffusion models may not perform as well as few-step (2–4 step) models (Salimans & Ho, 2022), we empirically find that with additional guidance, their performance improves significantly, as presented in Section 5.3. Second, as the guidance network $h_{\boldsymbol{\psi}}$ now becomes time-independent, we empirically observe that $h_{\boldsymbol{\psi}}$ is easy to train—with ten training epochs on the Pick-a-Pic V1 dataset, our guidance network produces high-quality images, which can be found in Section 5.2. We summarize the overall learning pipeline in Algorithm 1 in the Appendix.

## 4 Proposed Framework for Flow Matching

**Training-free Alignment Framework for Flow Matching.** Given that state-of-the-art text-to-image models are increasingly built on Diffusion Transformers (Peebles & Xie, 2022) and flow matching (Lipman et al., 2023), we also present the exact form of reward-guided velocity fields for flow matching. Similar to Section 3, we let the conditional distribution of the reference flow matching model $\pi_{\text{ref}}(\mathbf{x}_1|y)$ be denoted as distribution $p$, and let the reward-weighted distribution $\pi_{\text{r}}(\mathbf{x}_1|y)$ defined in Eq. (7) be denoted as distribution $q$. We summarize the guidance term in the following theorem.

**Theorem 4.1.** *Let $\boldsymbol{\phi}_q^*$ be the optimal solution for the conditional flow matching model trained on target domain $q(\mathbf{x}_1, y)$ (where $\mathbf{x}_1$ are sampled from data distribution, $\mathbf{v}_q(\mathbf{x}_t, y, t)$ denotes the oracle velocity field on target distribution), i.e., $\boldsymbol{\phi}_q^*$ equals*

$$\arg\min_{\boldsymbol{\phi}} \mathbb{E}_t \left\{ \mathbb{E}_{q_t(\mathbf{x}_t, y)} \left[ \left\| \mathbf{v}_{\boldsymbol{\phi}}(\mathbf{x}_t, y, t) - \mathbf{v}_q(\mathbf{x}_t, y, t) \right\|_2^2 \right] \right\},$$

*then*

$$\mathbf{v}_{\boldsymbol{\phi}_q^*}(\mathbf{x}_t, y, t) = \mathbf{v}_{\boldsymbol{\phi}_p}(\mathbf{x}_t, y, t) + \mathbb{E}_{\mathbf{x}_1 \sim p_{1|t}(\mathbf{x}_1|\mathbf{x}_t, y)} \left[ \left( R(\mathbf{x}_1, \mathbf{x}_t, y) - 1 \right) \boldsymbol{v}_t(\mathbf{x}_t \mid \mathbf{x}_1, y) \right], \tag{16}$$

*where*

$$R(\mathbf{x}_1, \mathbf{x}_t, y) = \frac{\exp\left(\frac{1}{\beta} r(\mathbf{x}_1, y)\right)}{\mathbb{E}_{\mathbf{x}_1' \sim p_{1|t}(\mathbf{x}_1|\mathbf{x}_t, y)} \left[\exp\left(\frac{1}{\beta} r(\mathbf{x}_1', y)\right)\right]}.$$

**Estimation of the Guidance Term for Flow Matching.** Different from the guidance term of diffusion models in Eq. (10), the guidance for flow matching in Eq. (16) does not have the adversarial problem. The guidance term is a conditional expectation without the gradient operator.

To enable fast sampling, we use importance sampling to convert the conditional expectation under $p(\mathbf{x}_1 \mid \mathbf{x}_t, y)$ into an expectation under $p(\mathbf{x}_1 \mid y)$. Following the derivation in Appendix A.2, we can calculate the guidance term of flow matching by

$$\mathbb{E}_{\mathbf{x}_1 \sim p(\mathbf{x}_1|y)} \left[ \left( \frac{\exp\left(\frac{1}{\beta} r(\mathbf{x}_1, y)\right)}{\mathbb{E}_{\mathbf{x}_1}\left[ \exp\left(\frac{1}{\beta} r(\mathbf{x}_1, y)\right) \frac{p_{t|1}(\mathbf{x}_t|\mathbf{x}_1,y)}{\mathbb{E}_{\mathbf{x}_1}[p_{t|1}(\mathbf{x}_t|\mathbf{x}_1,y)]} \right]} - 1 \right) \boldsymbol{v}_t(\mathbf{x}_t \mid \mathbf{x}_1, y) \frac{p_{t|1}(\mathbf{x}_t \mid \mathbf{x}_1, y)}{\mathbb{E}_{\mathbf{x}_1}[p_{t|1}(\mathbf{x}_t \mid \mathbf{x}_1, y)]} \right].$$

Therefore, the guidance term can be estimated by sampling from the marginal data distribution $p(\mathbf{x}_1 \mid y)$, instead of repeatedly sampling from the conditional distribution $p_{1|t}(\mathbf{x}_1 \mid \mathbf{x}_t, y)$. Compared with the finetuning-free diffusion method proposed in Eq. (13), this flow-matching formulation is training-free and offers greater computational efficiency.

## 5 Experimental Results

In this section, we present a comprehensive experimental evaluation, demonstrating the effectiveness of our two frameworks for sampling directly from reward-guided distributions. We first outline our experimental setup and evaluation criteria in Section 5.1, followed by benchmark results against state-of-the-art methods in Section 5.2. Finally, we provide an in-depth ablation study that validates our key theoretical claims and demonstrates the superior performance of our guidance network in Section 5.3.

### 5.1 Experimental Setup

For the experiments on diffusion models, we follow the official configurations recommended for SPO (Liang et al., 2025), Diffusion-DPO (Wallace et al., 2023), and MAPO (She et al., 2024). Diffusion-DPO and MAPO are fine-tuned on the Pick-a-Pic V2 dataset, which contains over 800k image preference pairs. In contrast, SPO is fine-tuned online using 4k text prompts (without images) randomly selected from Pick-a-Pic V1. Our method trains the guidance network offline using 583k image preference pairs from Pick-a-Pic V1. Overall, our method and the competing models in the text-to-image alignment benchmark are trained on comparable datasets, allowing for a fair comparison. We adopt Stable Diffusion XL (SDXL)-Turbo (Sauer et al., 2023) as the reference model for one-step text-to-image generation. For the experiments on flow matching, we adopt the state-of-the-art SD3.5 Large Turbo (Esser et al., 2024) as the backbone. The official recommendation for the number of sampling steps is four to eight, and we use four steps for all experiments.

**Implementation Details.** Since the guidance network takes noisy images $\mathbf{x}_T$ and prompts $y$ as input and outputs a scalar value, we adopt the same variational autoencoder (VAE), tokenizer, and text encoder from the reference diffusion model for encoding image and text. Consequently, the trainable parameters of our guidance network are quite small. In practice, we adopt two convolutional layers for processing VAE-encoded feature maps and a five-layer multi-layer perceptron (MLP) to project the image and text embedding to a scalar. The total parameter size of the guidance network is only 72 MB, making it lightweight and easy to train. We train the guidance network on the Pick-a-Pic training dataset for 10 epochs with batch size 32, Adam optimizer, learning rate 1e-3, and hyperparameters $\eta = 1$.

**Evaluation Criterion.** Following established evaluation protocols (Wallace et al., 2023; Liang et al., 2025), we report quantitative results using 500 validation prompts from the validation unique split of Pick-a-Pic. We adopt four evaluation criteria to evaluate different aspects of image quality. PickScore (Kirstain et al., 2023) measures overall human preference by aggregating judgments on aesthetic appeal, coherence, and realism. HPSV2 (Wu et al., 2023) assesses prompt adherence, ensuring the generated image accurately reflects the given textual description. ImageReward (Xu et al., 2023) quantifies human preference based on fine-grained attributes such as composition, detail preservation, and semantic relevance. Lastly, the aesthetic evaluation model from LAION (Schuhmann, 2022) focuses on visual appeal, capturing factors such as color harmony, style, and artistic quality.

Table 2: Benchmark comparison of different methods on text-to-image alignment. Results are grouped by base model.

| Type | Method | PickScore | HPSV2 | ImageReward | Aesthetic | Training GPU Hour |
|---|---|---|---|---|---|---|
| **Base Model: SDXL** | | | | | | |
| Baseline | SDXL | 21.95 | 26.95 | 0.5380 | 5.950 | – |
| Training-free | Direct backpropagate | 21.84 | 27.53 | 0.5870 | 5.922 | – |
| | Tweedie's formula | 22.34 | 28.76 | 0.9501 | 6.002 | – |
| Finetuning-based | Diff.-DPO | 22.64 | 29.31 | 0.9436 | 6.015 | 4800 |
| | SPO | 23.06 | 31.80 | **1.0803** | 6.364 | 234 |
| Finetuning-free | Ours | **23.08** | **32.12** | 1.0625 | **6.452** | **92** |
| **Base Model: SD3.5 Large Turbo** | | | | | | |
| Baseline | SD3.5 Large Turbo | 22.30 | 30.29 | 1.0159 | 6.5190 | – |
| Finetuning-free | Ours | **23.14** | **32.31** | **1.1025** | **6.5280** | – |

## 5.2 Experimental Results

As shown in Table 2, our method surpasses baseline approaches across four evaluation criteria, demonstrating the effectiveness of the two proposed frameworks in enhancing text-to-image alignment. The improvements are observed in both perceptual quality and semantic coherence, indicating that our guidance network successfully refines image generation to better match textual descriptions. This performance gain highlights the advantages of our lightweight architecture and the optimization strategy used during training. Figure 1 provides a qualitative comparison with baseline methods, further illustrating the superior visual fidelity and text alignment achieved by our approach.

## 5.3 Ablation study

In this section, we first verify the advantages of our proposed method against other finetuning-free guidance methods as summarized in Table 1. We then analyze the impact of few-step (2–4 step) generation compared to one-step generation, highlighting how our guidance term significantly enhances performance.

As illustrated in Figure 3 in the Appendix, vanilla guidance methods struggle to induce meaningful improvements in generated images, even with carefully tuned guidance strength. Increasing the guidance parameter $\alpha$ often leads to undesirable artifacts rather than quality improvements. In contrast, our method effectively enhances image generation by leveraging a regularized guidance network, demonstrating its ability to refine scene details and improve alignment with input prompts.

To further explore this, we examine the performance of our method against two vanilla guidance techniques, Tweedie's and Backpropagate, as well as the no guidance baseline, all under a one-step sampling condition. As shown in Table 3, our method achieves the highest PickScore. This demonstrates that our regularized guidance network provides a substantial improvement over no guidance scenario and traditional methods. Consistent with prior studies, increasing the number of steps from one to two or three results in improved image quality, as shown in Figure 3 and Table 3. However, our method enables one-step generation to achieve performance even better than 2- or 3-step generation, highlighting the power of our guidance network. In Appendix B, we include the sensitive analysis of the regularization strength.

# 6 Related Work

Existing alignment methods can be broadly categorized into two approaches: RLHF-based method that uses policy gradient to update the diffusion models and flow matching, and DPO-based methods that use a parametrization trick to update the diffusion models without explicitly learning the reward function.

Table 3: Ablation study comparing the performance of our method with no guidance and two vanilla guidance methods under one-step and multi-step generation. Our method outperforms all baselines, which demonstrates the effectiveness of our guidance network in refining image quality and prompt alignment.

| Method | PickScore |
|---|---|
| Ours (1 step) | **23.08** |
| No guidance (1 step) | 22.14 |
| Tweedie's (1 step) | 22.34 |
| Backpropagate (1 step) | 21.84 |
| No guidance (2 steps) | 22.64 |
| No guidance (3 steps) | 22.56 |

**RLHF-based alignment of diffusion model and flow matching.** Lee et al. (Lee et al., 2023) first train a reward model to predict human feedback and adopt a reward-weighted finetuning objective to align the diffusion model. In Fan et al. (2023); Black et al. (2024), diffusion models are updated using policy gradient algorithms under Kullback–Leibler (KL) constraints. Clark et al. (Clark et al., 2024) propagate gradients of the reward function through the full sampling procedure, and reduce memory costs by adopting low-rank adaptation (LoRA) (Hu et al., 2022) and gradient checkpointing (Chen et al., 2016). In Liu et al. (2025); Li et al. (2025); Xue et al. (2025); He et al. (2025), the authors improve GRPO (Shao et al., 2024) for the alignment of flow matching.

**DPO-based alignment of diffusion model.** A line of work (Wallace et al., 2023; Yang et al., 2023) directly applies DPO (Rafailov et al., 2024) to align the diffusion model with human preference. Liang et al. (Liang et al., 2025) propose a step-aware preference model and a step-wise resampler to align the preference optimization target with the denoising performance at each timestep. Yang et al. (Yang et al., 2024) take on a finer dense reward perspective and derive a tractable alignment objective that emphasizes the initial steps.

**Training-free guidance.** This line of work (Chung et al., 2023; Graikos et al., 2022; Lu et al., 2023; Song et al., 2023a; Bansal et al., 2023; Yu et al., 2023; Shen et al., 2024; Ye et al., 2024) explores the use of diffusion models as plug-and-play priors for solving inverse problems. Some work Shen et al. (2024); Tang et al. (2025); Uehara et al. (2024); Ma et al. (2025); Singhal et al. (2025) study inference-time optimization for alignment. However, to the best of our knowledge, there has been limited exploration of applying guidance on diffusion models to address the challenge of text-to-image alignment in the context of one-step generation. Also, there has been limited exploration of training-free guidance on flow matching of text-to-image alignment. This gap motivates our work.

**Plug-and-play control for text-to-image generation.** A related but distinct line of work keeps the pre-trained text-to-image model fixed and modifies the sampling process for controllability. These methods use attention control, feature injection, layout constraints, dense regional prompts, or initial-noise optimization (Hertz et al., 2023; Tumanyan et al., 2023; Chefer et al., 2023; Bar-Tal et al., 2023; Xie et al., 2023; Kim et al., 2023; Guo et al., 2024; Eyring et al., 2024). They show that pre-trained generators can be steered without updating the backbone model. However, most of them are designed for specific control signals rather than general reward alignment. In contrast, our method starts from the reward-weighted distribution induced by a specified reward function and derives the corresponding score or velocity guidance.

## 7 Conclusion

In this paper, we introduced two novel framework for aligning text-to-image diffusion models and flow matching models with human preferences. By formulating alignment as sampling from a reward-weighted distribution, our approach leverages a plug-and-play guidance mechanism. Specifically, we decomposed the score function (velocity field) of the reward-weighted distribution into the pre-trained score (velocity field) plus a reward-driven guidance term. For diffusion models, we identify that the adversarial nature of the guidance

can introduce undesirable artifacts, and we propose a finetuning-free approach that trains a lightweight guidance network to estimate the conditional expectation of the reward, together with a regularization strategy that stabilizes the guidance landscape. Empirically, our method achieves performance comparable to finetuning-based approaches for one-step generation while reducing computational cost by at least 60%. For flow matching, we derive the exact form of velocity guidance and propose a training-free estimator that improves generation quality without additional training.

## Broader Impact Statement

We propose an efficient alignment framework for text-to-image generative models that significantly reduces computational costs, making high-quality aligned generation more accessible. Like all advances in generative modeling, this work could lower the barrier to creating misleading visual content, including Deepfakes for disinformation. We advocate for the parallel development of robust detection tools and governance mechanisms to mitigate potential misuse.

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

# A    Theoretical Details for Section 3 and Section 4

## A.1    Proof of Theorem 3.1

We first restate the complete theorem as follows:

**Theorem A.1.** *Let the conditional distribution of reference diffusion model $\pi_{ref}(\mathbf{x}|y)$ be denoted as distribution $p$ and the reward-weighted distribution $\pi_r(\mathbf{x}|y)$ defined in Eq. (7) as distribution $q$. Assume $\mathbf{x}_t$ and $y$ are conditionally independent given $\mathbf{x}_0$ in the forward process, i.e., $p(\mathbf{x}_t|\mathbf{x}_0, y) = p(\mathbf{x}_t|\mathbf{x}_0)$, $\forall t \in [0, T]$. Additionally, assume the forward process on the reward-weighted distribution is identical to that on the reference distribution $q(\mathbf{x}_t|\mathbf{x}_0) = p(\mathbf{x}_t|\mathbf{x}_0)^4$, and $\phi^*$ is the optimal solution for the conditional diffusion model trained on target domain $q(\mathbf{x}_0, y)$, i.e.,*

$$\phi^* = \arg\min_{\phi} \mathbb{E}_t\left\{\lambda(t)\mathbb{E}_{q_t(\mathbf{x}_t,y)}\left[\left\|\mathbf{s}_\phi(\mathbf{x}_t, y, t) - \nabla_{\mathbf{x}_t}\log q_t(\mathbf{x}_t|y)\right\|_2^2\right]\right\}, \tag{17}$$

*then*

$$\mathbf{s}_{\phi^*}(\mathbf{x}_t, y, t) = \underbrace{\nabla_{\mathbf{x}_t}\log p_t(\mathbf{x}_t|y)}_{\substack{pre\text{-}trained\ conditional\ model \\ on\ source}} + \underbrace{\nabla_{\mathbf{x}_t}\log \mathbb{E}_{p(\mathbf{x}_0|\mathbf{x}_t,y)}\left[\exp(\frac{1}{\beta}r(\mathbf{x}_0, y))\right]}_{conditional\ guidance}. \tag{18}$$

*Proof.* The proof is based on the theoretical framework of Ouyang et al. (2024). For the ease of readers, we incorporate the relevant conclusion from their work as lemmas below. To prove Eq. (18), we first build the connection between the Conditional Score Matching on the target domain and Importance Weighted Conditional Denoising Score Matching on the source domain in the following Lemma:

**Lemma A.2.** *Conditional Score Matching on the target domain is equivalent to Importance Weighted Denoising Score Matching on the source domain, i.e.,*

$$\begin{aligned}
\phi^* =& \arg\min_{\phi} \mathbb{E}_t\left\{\lambda(t)\mathbb{E}_{q_t(\mathbf{x}_t,y)}\left[\|\mathbf{s}_\phi(\mathbf{x}_t, y, t) - \nabla_{\mathbf{x}_t}\log q_t(\mathbf{x}_t|y)\|_2^2\right]\right\} \\
=& \arg\min_{\phi} \mathbb{E}_t\left\{\lambda(t)\mathbb{E}_{p(\mathbf{x}_0,y)}\mathbb{E}_{p(\mathbf{x}_t|\mathbf{x}_0)}\left[\|\mathbf{s}_\phi(\mathbf{x}_t, y, t) - \nabla_{\mathbf{x}_t}\log p(\mathbf{x}_t|\mathbf{x}_0)\|_2^2 \frac{q(\mathbf{x}_0, y)}{p(\mathbf{x}_0, y)}\right]\right\}.
\end{aligned}$$

*Proof of Lemma A.2.* We first connect the Conditional Score Matching objective in the target domain to the Conditional Denoising Score Matching objective in target distribution, which is proven by Batzolis et al. (2021), i.e.,

$$\begin{aligned}
\phi^* =& \arg\min_{\phi} \mathbb{E}_t\left\{\lambda(t)\mathbb{E}_{q_t(\mathbf{x}_t,y)}\left[\|\mathbf{s}_\phi(\mathbf{x}_t, y, t) - \nabla_{\mathbf{x}_t}\log q_t(\mathbf{x}_t|y)\|_2^2\right]\right\} \\
=& \arg\min_{\phi} \mathbb{E}_t\left\{\lambda(t)\mathbb{E}_{q(\mathbf{x}_0,y)}\mathbb{E}_{q(\mathbf{x}_t|\mathbf{x}_0)}\left[\|\mathbf{s}_\phi(\mathbf{x}_t, y, t) - \nabla_{\mathbf{x}_t}\log q(\mathbf{x}_t|\mathbf{x}_0)\|_2^2\right]\right\}.
\end{aligned}$$

Then we split the mean squared error of the Conditional Denoising Score Matching objective on the target distribution into three terms as follows:

$$\begin{aligned}
&\mathbb{E}_{q(\mathbf{x}_0,y)}\mathbb{E}_{q(\mathbf{x}_t|\mathbf{x}_0)}\left[\|\mathbf{s}_\phi(\mathbf{x}_t, y, t) - \nabla_{\mathbf{x}_t}\log q(\mathbf{x}_t|\mathbf{x}_0)\|_2^2\right] \\
=&\mathbb{E}_{q(\mathbf{x}_0,\mathbf{x}_t,y)}\left[\|\mathbf{s}_\phi(\mathbf{x}_t, y, t)\|_2^2\right] - 2\mathbb{E}_{q(\mathbf{x}_0,\mathbf{x}_t,y)}\left[\langle\mathbf{s}_\phi(\mathbf{x}_t, y, t), \nabla_{\mathbf{x}_t}\log q(\mathbf{x}_t|\mathbf{x}_0)\rangle\right] + C_1, \tag{19}
\end{aligned}$$

where $C_1 = \mathbb{E}_{q(\mathbf{x}_0,\mathbf{x}_t,y)}\left[\|\nabla_{\mathbf{x}_t}\log q(\mathbf{x}_t|\mathbf{x}_0)\|_2^2\right]$ is a constant independent with $\phi$, and $q(\mathbf{x}_t|\mathbf{x}_0, y) = q(\mathbf{x}_t|\mathbf{x}_0)$ because of conditional independent of $\mathbf{x}_t$ and $y$ given $\mathbf{x}_0$ by assumption. We can similarly split the mean

---

[4]These two assumptions are mild since $\mathbf{x}_0$ contains all information about $y$ and $p(\mathbf{x}_t|\mathbf{x}_0)$ and $q(\mathbf{x}_t|\mathbf{x}_0)$ are forward noising process, which is easy to control.

squared error of Denoising Score Matching on the source domain into three terms as follows:

$$\mathbb{E}_{p(\mathbf{x}_0,y)}\mathbb{E}_{p(\mathbf{x}_t|\mathbf{x}_0)}\left[\|\mathbf{s}_\phi(\mathbf{x}_t,y,t)-\nabla_{\mathbf{x}_t}\log p(\mathbf{x}_t|\mathbf{x}_0)\|_2^2\frac{q(\mathbf{x}_0,y)}{p(\mathbf{x}_0,y)}\right]$$

$$=\mathbb{E}_{p(\mathbf{x}_0,\mathbf{x}_t,y)}\left[\|\mathbf{s}_\phi(\mathbf{x}_t,y,t)\|_2^2\frac{q(\mathbf{x}_0,y)}{p(\mathbf{x}_0,y)}\right]-2\mathbb{E}_{p(\mathbf{x}_0,\mathbf{x}_t,y)}\left[\langle\mathbf{s}_\phi(\mathbf{x}_t,y,t),\nabla_{\mathbf{x}_t}\log p(\mathbf{x}_t|\mathbf{x}_0)\rangle\frac{q(\mathbf{x}_0,y)}{p(\mathbf{x}_0,y)}\right] \quad (20)$$

$$+\,C_2,$$

where $C_2$ is a constant independent with $\phi$.

It is obvious to show that the first term in Eq. (19) is equal to the first term in Eq. (20), i.e.,

$$\mathbb{E}_{p(\mathbf{x}_0,\mathbf{x}_t,y)}\left[\|\mathbf{s}_\phi(\mathbf{x}_t,y,t)\|_2^2\frac{q(\mathbf{x}_0,y)}{p(\mathbf{x}_0,y)}\right]$$

$$=\int_{\mathbf{x}_0}\int_{\mathbf{x}_t}\int_y p(\mathbf{x}_0,y)p(\mathbf{x}_t|\mathbf{x}_0)\|\mathbf{s}_\phi(\mathbf{x}_t,y,t)\|_2^2\frac{q(\mathbf{x}_0,y)}{p(\mathbf{x}_0,y)}d\mathbf{x}_0 d\mathbf{x}_t dy$$

$$=\int_{\mathbf{x}_0}\int_{\mathbf{x}_t}\int_y p(\mathbf{x}_0,y)q(\mathbf{x}_t|\mathbf{x}_0)\|\mathbf{s}_\phi(\mathbf{x}_t,y,t)\|_2^2\frac{q(\mathbf{x}_0,y)}{p(\mathbf{x}_0,y)}d\mathbf{x}_0 d\mathbf{x}_t dy$$

$$=\int_{\mathbf{x}_0}\int_{\mathbf{x}_t}\int_y q(\mathbf{x}_0,\mathbf{x}_t,y)\|\mathbf{s}_\phi(\mathbf{x}_t,y,t)\|_2^2 d\mathbf{x}_0 d\mathbf{x}_t dy$$

$$=\mathbb{E}_{q(\mathbf{x}_0,\mathbf{x}_t,y)}\left[\|\mathbf{s}_\phi(\mathbf{x}_t,y,t)\|_2^2\right].$$

And the second term is also equivalent:

$$\mathbb{E}_{p(\mathbf{x}_0,\mathbf{x}_t,y)}\left[\langle\mathbf{s}_\phi(\mathbf{x}_t,y,t),\nabla_{\mathbf{x}_t}\log p(\mathbf{x}_t|\mathbf{x}_0)\rangle\frac{q(\mathbf{x}_0,y)}{p(\mathbf{x}_0,y)}\right]$$

$$=\int_{\mathbf{x}_0}\int_{\mathbf{x}_t}\int_y p(\mathbf{x}_0,\mathbf{x}_t,y)\langle\mathbf{s}_\phi(\mathbf{x}_t,y,t),\frac{\nabla_{\mathbf{x}_t}p(\mathbf{x}_t|\mathbf{x}_0)}{p(\mathbf{x}_t|\mathbf{x}_0)}\rangle\frac{q(\mathbf{x}_0,y)}{p(\mathbf{x}_0,y)}d\mathbf{x}_0 d\mathbf{x}_t dy$$

$$=\int_{\mathbf{x}_0}\int_{\mathbf{x}_t}\int_y p(\mathbf{x}_0,\mathbf{x}_t,y)\langle\mathbf{s}_\phi(\mathbf{x}_t,y,t),\frac{\nabla_{\mathbf{x}_t}q(\mathbf{x}_t|\mathbf{x}_0)}{p(\mathbf{x}_t|\mathbf{x}_0)}\rangle\frac{q(\mathbf{x}_0,y)}{p(\mathbf{x}_0,y)}d\mathbf{x}_0 d\mathbf{x}_t dy$$

$$=\int_{\mathbf{x}_0}\int_{\mathbf{x}_t}\int_y\langle\mathbf{s}_\phi(\mathbf{x}_t,y,t),\nabla_{\mathbf{x}_t}q(\mathbf{x}_t|\mathbf{x}_0)\rangle q(\mathbf{x}_0,y)d\mathbf{x}_0 d\mathbf{x}_t dy$$

$$=\int_{\mathbf{x}_0}\int_{\mathbf{x}_t}\int_y\langle\mathbf{s}_\phi(\mathbf{x}_t,y,t),\nabla_{\mathbf{x}_t}\log q(\mathbf{x}_t|\mathbf{x}_0)\rangle q(\mathbf{x}_t|\mathbf{x}_0)q(\mathbf{x}_0,y)d\mathbf{x}_0 d\mathbf{x}_t dy$$

$$=\mathbb{E}_{q(\mathbf{x}_0,\mathbf{x}_t,y)}\left[\langle\mathbf{s}_\phi(\mathbf{x}_t,y,t),\nabla_{\mathbf{x}_t}\log q(\mathbf{x}_t|\mathbf{x}_0)\rangle\right].$$

$$\square$$

**Lemma A.3.** *Assume $\mathbf{x}_t$ and $y$ are conditional independent given $\mathbf{x}_0$ in the forward process, i.e., $p(\mathbf{x}_t|\mathbf{x}_0,y) = p(\mathbf{x}_t|\mathbf{x}_0)$, $\forall t \in [0,T]$, and let the forward process on the target domain be identical to that on the source domain $q(\mathbf{x}_t|\mathbf{x}_0) = p(\mathbf{x}_t|\mathbf{x}_0)$, and $\phi^*$ is the optimal solution for the conditional diffusion model trained on target domain $q(\mathbf{x}_0,y)$, i.e.,*

$$\phi^* = \arg\min_\phi \mathbb{E}_t\left\{\lambda(t)\mathbb{E}_{q_t(\mathbf{x}_t,y)}\left[\|\mathbf{s}_\phi(\mathbf{x}_t,y,t)-\nabla_{\mathbf{x}_t}\log q_t(\mathbf{x}_t|y)\|_2^2\right]\right\}, \quad (21)$$

*then*

$$\mathbf{s}_{\phi^*}(\mathbf{x}_t,y,t) = \nabla_{\mathbf{x}_t}\log p_t(\mathbf{x}_t|y) + \nabla_{\mathbf{x}_t}\log\mathbb{E}_{p(\mathbf{x}_0|\mathbf{x}_t,y)}\left[\frac{q(\mathbf{x}_0,y)}{p(\mathbf{x}_0,y)}\right]. \quad (22)$$

*Proof of Lemma A.3.* According to Lemma A.2, the optimal solution satisfies

$$\phi^* = \arg\min_\phi \mathbb{E}_t\left\{\lambda(t)\mathbb{E}_{p(\mathbf{x}_0,y)}\mathbb{E}_{p(\mathbf{x}_t|\mathbf{x}_0)}\left[\|\mathbf{s}_\phi(\mathbf{x}_t,y,t)-\nabla_{\mathbf{x}_t}\log p(\mathbf{x}_t|\mathbf{x}_0)\|_2^2\frac{q(\mathbf{x}_0,y)}{p(\mathbf{x}_0,y)}\right]\right\}$$

where $Z(y) = \int p(\mathbf{x}_0, y) \exp\left(\frac{1}{\beta}r(\mathbf{x}_0, y)\right) \mathrm{d}\mathbf{x}$. Then, we use Importance Weighted Conditional Denoising Score Matching on the source domain to get the analytic form of $\mathbf{s}_{\boldsymbol{\phi}^*}$ as follows:

$$\mathbf{s}_{\boldsymbol{\phi}^*}(\mathbf{x}_t, y, t) = \frac{\mathbb{E}_{p(\mathbf{x}_0|\mathbf{x}_t, y)}\left[\nabla_{\mathbf{x}_t} \log p(\mathbf{x}_t|\mathbf{x}_0)\frac{q(\mathbf{x}_0, y)}{p(\mathbf{x}_0, y)}\right]}{\mathbb{E}_{p(\mathbf{x}_0|\mathbf{x}_t, y)}\left[\frac{q(\mathbf{x}_0, y)}{p(\mathbf{x}_0, y)}\right]}.$$

Moreover, the RHS of Eq. (22) can be rewritten as:

$$\begin{aligned}
\mathrm{RHS} =&\nabla_{\mathbf{x}_t} \log p_t(\mathbf{x}_t|y) + \nabla_{\mathbf{x}_t} \log \mathbb{E}_{p(\mathbf{x}_0|\mathbf{x}_t, y)}\left[\frac{q(\mathbf{x}_0, y)}{p(\mathbf{x}_0, y)}\right] \\
=&\nabla_{\mathbf{x}_t} \log p_t(\mathbf{x}_t|y) + \frac{\nabla_{\mathbf{x}_t}\mathbb{E}_{p(\mathbf{x}_0|\mathbf{x}_t, y)}\left[\frac{q(\mathbf{x}_0, y)}{p(\mathbf{x}_0, y)}\right]}{\mathbb{E}_{p(\mathbf{x}_0|\mathbf{x}_t, y)}\left[\frac{q(\mathbf{x}_0, y)}{p(\mathbf{x}_0, y)}\right]} \\
=&\nabla_{\mathbf{x}_t} \log p_t(\mathbf{x}_t|y) + \frac{\mathbb{E}_{p(\mathbf{x}_0|\mathbf{x}_t, y)}\left[\frac{q(\mathbf{x}_0, y)}{p(\mathbf{x}_0, y)}\nabla_{\mathbf{x}_t} \log p(\mathbf{x}_0|\mathbf{x}_t, y)\right]}{\mathbb{E}_{p(\mathbf{x}_0|\mathbf{x}_t, y)}\left[\frac{q(\mathbf{x}_0, y)}{p(\mathbf{x}_0, y)}\right]}.
\end{aligned}$$

Since

$$\begin{aligned}
\nabla_{\mathbf{x}_t} \log p(\mathbf{x}_0|\mathbf{x}_t, y) &= \nabla_{\mathbf{x}_t} \log p(\mathbf{x}_t|\mathbf{x}_0, y) + \nabla_{\mathbf{x}_t} \log p(\mathbf{x}_0|y) - \nabla_{\mathbf{x}_t} \log p_t(\mathbf{x}_t|y) \\
&= \nabla_{\mathbf{x}_t} \log p(\mathbf{x}_t|\mathbf{x}_0, y) - \nabla_{\mathbf{x}_t} \log p_t(\mathbf{x}_t|y), \\
&= \nabla_{\mathbf{x}_t} \log p(\mathbf{x}_t|\mathbf{x}_0) - \nabla_{\mathbf{x}_t} \log p_t(\mathbf{x}_t|y),
\end{aligned}$$

we can further simplify the RHS of Eq. (22) as follows:

$$\begin{aligned}
\mathrm{RHS} =&\nabla_{\mathbf{x}_t} \log p_t(\mathbf{x}_t|y) + \frac{\mathbb{E}_{p(\mathbf{x}_0|\mathbf{x}_t, y)}\left[\frac{q(\mathbf{x}_0, y)}{p(\mathbf{x}_0, y)}\nabla_{\mathbf{x}_t} \log p(\mathbf{x}_t|\mathbf{x}_0)\right]}{\mathbb{E}_{p(\mathbf{x}_0|\mathbf{x}_t, y)}\left[\frac{q(\mathbf{x}_0, y)}{p(\mathbf{x}_0, y)}\right]} - \nabla_{\mathbf{x}_t} \log p_t(\mathbf{x}_t|y) \\
=&\frac{\mathbb{E}_{p(\mathbf{x}_0|\mathbf{x}_t, y)}\left[\nabla_{\mathbf{x}_t} \log p(\mathbf{x}_t|\mathbf{x}_0)\frac{q(\mathbf{x}_0, y)}{p(\mathbf{x}_0, y)}\right]}{\mathbb{E}_{p(\mathbf{x}_0|\mathbf{x}_t, y)}\left[\frac{q(\mathbf{x}_0, y)}{p(\mathbf{x}_0, y)}\right]} \\
=&\mathbf{s}_{\boldsymbol{\phi}^*}(\mathbf{x}_t, t).
\end{aligned}$$

Thereby, we finish the proof. $\qquad\square$

According to the lemma A.3, we replace the density ratio $\frac{q(\mathbf{x}_0, y)}{p(\mathbf{x}_0, y)}$ by $\frac{\exp\left(\frac{1}{\beta}r(\mathbf{x}_0, y)\right)}{Z(y)}$, we get

$$\begin{aligned}
\mathbf{s}_{\boldsymbol{\phi}^*}(\mathbf{x}_t, y, t) &= \nabla_{\mathbf{x}_t} \log p_t(\mathbf{x}_t|y) + \nabla_{\mathbf{x}_t} \log \mathbb{E}_{p(\mathbf{x}_0|\mathbf{x}_t, y)}\left[\frac{q(\mathbf{x}_0, y)}{p(\mathbf{x}_0, y)}\right] \\
&= \nabla_{\mathbf{x}_t} \log p_t(\mathbf{x}_t|y) + \nabla_{\mathbf{x}_t} \log \mathbb{E}_{p(\mathbf{x}_0|\mathbf{x}_t, y)}\left[\frac{\exp\left(\frac{1}{\beta}r(\mathbf{x}_0, y)\right)}{Z(y)}\right] \\
&= \nabla_{\mathbf{x}_t} \log p_t(\mathbf{x}_t|y) + \nabla_{\mathbf{x}_t} \log \mathbb{E}_{p(\mathbf{x}_0|\mathbf{x}_t, y)}\left[\exp\left(\frac{1}{\beta}r(\mathbf{x}_0, y)\right)\right]
\end{aligned}$$

Thereby, we finish the proof. $\qquad\square$

## A.2 Proof of Theorem 4.1

We provide a detailed discussion about training-free guidance of flow matching in this subsection.

*Proof of Theorem 4.1.* Denote $\boldsymbol{v}_t(\mathbf{x}_t, y)$ and $\boldsymbol{v}_t(\mathbf{x}_t \mid \mathbf{x}_1, y)$ as the marginal and conditional velocities, respectively. Then we have

$$\boldsymbol{v}_t^q(\mathbf{x}_t, y) = \mathbb{E}_{\mathbf{x}_1 \sim q_{1|t}(\mathbf{x}_1 | \mathbf{x}_t, y)} \left[ \boldsymbol{v}_t(\mathbf{x}_t \mid \mathbf{x}_1, y) \right]$$

$$= \mathbb{E}_{\mathbf{x}_1 \sim p_{1|t}(\mathbf{x}_1 | \mathbf{x}_t, y)} \left[ \boldsymbol{v}_t(\mathbf{x}_t \mid \mathbf{x}_1, y) \frac{q_{1|t}(\mathbf{x}_1 \mid \mathbf{x}_t, y)}{p_{1|t}(\mathbf{x}_1 \mid \mathbf{x}_t, y)} \right]$$

$$= \mathbb{E}_{\mathbf{x}_1 \sim p_{1|t}(\mathbf{x}_1 | \mathbf{x}_t, y)} \left[ \boldsymbol{v}_t(\mathbf{x}_t \mid \mathbf{x}_1, y) \frac{\frac{q_{t|1}(\mathbf{x}_t | \mathbf{x}_1, y) \, q_1(\mathbf{x}_1)}{q_t(\mathbf{x}_t, y)}}{\frac{p_{t|1}(\mathbf{x}_t | \mathbf{x}_1, y) \, p_1(\mathbf{x}_1)}{p_t(\mathbf{x}_t, y)}} \right]$$

$$= \mathbb{E}_{\mathbf{x}_1 \sim p_{1|t}(\mathbf{x}_1 | \mathbf{x}_t, y)} \left[ \boldsymbol{v}_t(\mathbf{x}_t \mid \mathbf{x}_1, y) \frac{q_{t|1}(\mathbf{x}_t \mid \mathbf{x}_1, y) \, q_1(\mathbf{x}_1) \, p_t(\mathbf{x}_t, y)}{p_{t|1}(\mathbf{x}_t \mid \mathbf{x}_1, y) \, p_1(\mathbf{x}_1) \, q_t(\mathbf{x}_t, y)} \right]$$

$$= \mathbb{E}_{\mathbf{x}_1 \sim p_{1|t}(\mathbf{x}_1 | \mathbf{x}_t, y)} \left[ \boldsymbol{v}_t(\mathbf{x}_t \mid \mathbf{x}_1, y) \frac{q_1(\mathbf{x}_1)}{p_1(\mathbf{x}_1)} \cdot \frac{p_t(\mathbf{x}_t, y)}{q_t(\mathbf{x}_t, y)} \right] \qquad \text{(because } q_{t|1}(\mathbf{x}_t \mid \mathbf{x}_1, y) = p_{t|1}(\mathbf{x}_t \mid \mathbf{x}_1, y))$$

$$= \mathbb{E}_{\mathbf{x}_1 \sim p_{1|t}(\mathbf{x}_1 | \mathbf{x}_t, y)} \left[ \boldsymbol{v}_t(\mathbf{x}_t \mid \mathbf{x}_1, y) \frac{\frac{q_1(\mathbf{x}_1)}{p_1(\mathbf{x}_1)}}{\frac{q_t(\mathbf{x}_t, y)}{p_t(\mathbf{x}_t, y)}} \right]$$

$$= \mathbb{E}_{\mathbf{x}_1 \sim p_{1|t}(\mathbf{x}_1 | \mathbf{x}_t, y)} \left[ \boldsymbol{v}_t(\mathbf{x}_t \mid \mathbf{x}_1, y) \frac{\frac{q_1(\mathbf{x}_1)}{p_1(\mathbf{x}_1)}}{\sum_{\mathbf{x}_1'} p_{1|t}(\mathbf{x}_1' \mid \mathbf{x}_t, y) \frac{q_1(\mathbf{x}_1')}{p_1(\mathbf{x}_1')}} \right]$$

$$= \mathbb{E}_{\mathbf{x}_1 \sim p_{1|t}(\mathbf{x}_1 | \mathbf{x}_t, y)} \left[ \boldsymbol{v}_t(\mathbf{x}_t \mid \mathbf{x}_1, y) \frac{\frac{q_1(\mathbf{x}_1)}{p_1(\mathbf{x}_1)}}{\mathbb{E}_{\mathbf{x}_1' \sim p_{1|t}(\mathbf{x}_1 | \mathbf{x}_t, y)} \left[ \frac{q_1(\mathbf{x}_1')}{p_1(\mathbf{x}_1')} \right]} \right]$$

$$= \mathbb{E}_{\mathbf{x}_1 \sim p_{1|t}(\mathbf{x}_1 | \mathbf{x}_t, y)} \left[ \boldsymbol{v}_t(\mathbf{x}_t \mid \mathbf{x}_1, y) \frac{\exp\left( \frac{1}{\beta} r(\mathbf{x}_1, y) \right)}{\mathbb{E}_{\mathbf{x}_1' \sim p_{1|t}(\mathbf{x}_1 | \mathbf{x}_t, y)} \left[ \exp\left( \frac{1}{\beta} r(\mathbf{x}_1', y) \right) \right]} \right]$$

$$= \mathbb{E}_{\mathbf{x}_1 \sim p_{1|t}(\mathbf{x}_1 | \mathbf{x}_t, y)} \left[ \boldsymbol{v}_t(\mathbf{x}_t \mid \mathbf{x}_1, y) \frac{\exp\left( \frac{1}{\beta} r(\mathbf{x}_1, y) \right)}{\mathbb{E}_{\mathbf{x}_1' \sim p_{1|t}(\mathbf{x}_1 | \mathbf{x}_t, y)} \left[ \exp\left( \frac{1}{\beta} r(\mathbf{x}_1', y) \right) \right]} \right]$$

$$= \boldsymbol{v}_t^p(\mathbf{x}_t, y) + \mathbb{E}_{\mathbf{x}_1 \sim p_{1|t}(\mathbf{x}_1 | \mathbf{x}_t, y)} \left[ \left( \frac{\exp\left( \frac{1}{\beta} r(\mathbf{x}_1, y) \right)}{\mathbb{E}_{\mathbf{x}_1' \sim p_{1|t}(\mathbf{x}_1 | \mathbf{x}_t, y)} \left[ \exp\left( \frac{1}{\beta} r(\mathbf{x}_1', y) \right) \right]} - 1 \right) \boldsymbol{v}_t(\mathbf{x}_t \mid \mathbf{x}_1, y) \right].$$

The above derivation is the training-based guidance for flow matching, where we need to train the first guidance network $\boldsymbol{\psi}_1^*$ satisfies:

$$h_{\boldsymbol{\psi}_1^*}(\mathbf{x}_t, y, t) = \mathbb{E}_{\mathbf{x}_1 \sim p_{1|t}(\mathbf{x}_1 | \mathbf{x}_t, y)} \left[ \exp\left( \frac{1}{\beta} r(\mathbf{x}_1, y) \right) \right]$$

by minimizing the objective

$$\mathcal{L}_{\text{guidance}}(\boldsymbol{\psi}_1) := \mathbb{E}_{p(\mathbf{x}_1, \mathbf{x}_t, y)} \left[ \left\| h_{\boldsymbol{\psi}_1}(\mathbf{x}_t, y, t) - \exp(\frac{1}{\beta} r(\mathbf{x}_1, y)) \right\|_2^2 \right].$$

And then we need the second guidance network $\boldsymbol{\psi}_2^*$ satisfies:

$$h_{\boldsymbol{\psi}_2^*}(\mathbf{x}_t, y, t) = \mathbb{E}_{\mathbf{x}_1 \sim p_{1|t}(\mathbf{x}_1 | \mathbf{x}_t, y)} \left[ \left( \frac{\exp\left(\frac{1}{\beta} r(\mathbf{x}_1, y)\right)}{\mathbb{E}_{\mathbf{x}_1' \sim p_{1|t}(\mathbf{x}_1 | \mathbf{x}_t, y)}\left[\exp\left(\frac{1}{\beta} r(\mathbf{x}_1', y)\right)\right]} - 1 \right) \boldsymbol{v}_t(\mathbf{x}_t | \mathbf{x}_1, y) \right]$$

by minimizing the objective

$$\mathcal{L}_{\text{guidance}}(\boldsymbol{\psi}_2) := \mathbb{E}_{p(\mathbf{x}_1, \mathbf{x}_t, y)} \left[ \left\| h_{\boldsymbol{\psi}_2}(\mathbf{x}_t, y, t) - \left( \frac{\exp\left(\frac{1}{\beta} r(\mathbf{x}_1, y)\right)}{h_{\boldsymbol{\psi}_1}(\mathbf{x}_t, y, t)} - 1 \right) \boldsymbol{v}_t(\mathbf{x}_t | \mathbf{x}_1, y) \right\|_2^2 \right].$$

The guidance network for flow matching is more complex than that used in diffusion models. The estimation errors from two guidance networks may accumulate and ultimately degrade generation performance. To address this limitation, we propose a training-free guidance method for flow matching that mitigates these issues.

$$\boldsymbol{v}_t^q(\mathbf{x}_t, y)$$

$$= \boldsymbol{v}_t^p(\mathbf{x}_t, y) + \mathbb{E}_{\mathbf{x}_1 \sim p_{1|t}(\mathbf{x}_1 | \mathbf{x}_t, y)} \left[ \left( \frac{\exp\left(\frac{1}{\beta} r(\mathbf{x}_1, y)\right)}{\mathbb{E}_{\mathbf{x}_1' \sim p_{1|t}(\mathbf{x}_1' | \mathbf{x}_t, y)}\left[\exp\left(\frac{1}{\beta} r(\mathbf{x}_1', y)\right)\right]} - 1 \right) \boldsymbol{v}_t(\mathbf{x}_t | \mathbf{x}_1, y) \right]$$

$$= \boldsymbol{v}_t^p(\mathbf{x}_t, y) + \int_{\mathbf{x}_1} \left( \frac{\exp\left(\frac{1}{\beta} r(\mathbf{x}_1, y)\right)}{\mathbb{E}_{\mathbf{x}_1' \sim p_{1|t}}\left[\exp\left(\frac{1}{\beta} r(\mathbf{x}_1', y)\right)\right]} - 1 \right) \boldsymbol{v}_t(\mathbf{x}_t | \mathbf{x}_1, y) \, p_{1|t}(\mathbf{x}_1 | \mathbf{x}_t, y) \, d\mathbf{x}_1$$

$$= \boldsymbol{v}_t^p(\mathbf{x}_t, y) + \int_{\mathbf{x}_1} \left( \frac{\exp\left(\frac{1}{\beta} r(\mathbf{x}_1, y)\right)}{\mathbb{E}_{\mathbf{x}_1 \sim p_{1|t}}\left[\exp\left(\frac{1}{\beta} r(\mathbf{x}_1, y)\right)\right]} - 1 \right) \boldsymbol{v}_t(\mathbf{x}_t | \mathbf{x}_1, y) \frac{p_{t|1}(\mathbf{x}_t | \mathbf{x}_1, y) \, p(\mathbf{x}_1 | y)}{p_t(\mathbf{x}_t | y)} d\mathbf{x}_1$$

$$= \boldsymbol{v}_t^p(\mathbf{x}_t, y) + \mathbb{E}_{\mathbf{x}_1 \sim p(\mathbf{x}_1 | y)} \left[ \left( \frac{\exp\left(\frac{1}{\beta} r(\mathbf{x}_1, y)\right)}{\mathbb{E}_{\mathbf{x}_1 \sim p_{1|t}}\left[\exp\left(\frac{1}{\beta} r(\mathbf{x}_1, y)\right)\right]} - 1 \right) \boldsymbol{v}_t(\mathbf{x}_t | \mathbf{x}_1, y) \frac{p_{t|1}(\mathbf{x}_t | \mathbf{x}_1, y)}{p_t(\mathbf{x}_t | y)} \right]$$

$$= \boldsymbol{v}_t^p(\mathbf{x}_t, y) + \mathbb{E}_{\mathbf{x}_1 \sim p(\mathbf{x}_1 | y)} \left[ \left( \frac{\exp\left(\frac{1}{\beta} r(\mathbf{x}_1, y)\right)}{\mathbb{E}_{\mathbf{x}_1 \sim p_{1|t}}\left[\exp\left(\frac{1}{\beta} r(\mathbf{x}_0, y)\right)\right]} - 1 \right) \boldsymbol{v}_t(\mathbf{x}_t | \mathbf{x}_1, y) \frac{p_{t|1}(\mathbf{x}_t | \mathbf{x}_1, y)}{\mathbb{E}_{\mathbf{x}_1 \sim p(\mathbf{x}_1 | y)}\left[p_{t|1}(\mathbf{x}_t | \mathbf{x}_1, y)\right]} \right]$$

$$= \boldsymbol{v}_t^p(\mathbf{x}_t, y) + \mathbb{E}_{\mathbf{x}_1 \sim p(\mathbf{x}_1 | y)} \left[ \left( \frac{\exp\left(\frac{1}{\beta} r(\mathbf{x}_1, y)\right)}{\mathbb{E}_{\mathbf{x}_1 \sim p(\mathbf{x}_1 | y)}\left[\exp\left(\frac{1}{\beta} r(\mathbf{x}_1, y)\right) \frac{p_{t|1}(\mathbf{x}_t | \mathbf{x}_1, y)}{\mathbb{E}_{\mathbf{x}_1 \sim p(\mathbf{x}_1 | y)}\left[p_{t|1}(\mathbf{x}_t | \mathbf{x}_1, y)\right]}\right]} - 1 \right) \right.$$
$$\left. \boldsymbol{v}_t(\mathbf{x}_t | \mathbf{x}_1, y) \frac{p_{t|1}(\mathbf{x}_t | \mathbf{x}_1, y)}{\mathbb{E}_{\mathbf{x}_1 \sim p(\mathbf{x}_1 | y)}\left[p_{t|1}(\mathbf{x}_t | \mathbf{x}_1, y)\right]} \right].$$

$\square$

### A.3 Proof of Lemma 3.2

*Proof.* The proof is straightforward and we include it below for completeness. Note that the objective function can be rewritten as

$$\mathcal{L}_{\text{guidance}}(\boldsymbol{\psi})$$

$$:=\mathbb{E}_{p(\mathbf{x}_0,\mathbf{x}_t,y)}\left[\left\|h_{\boldsymbol{\psi}}\left(\mathbf{x}_t,y,t\right)-\exp\left(\frac{1}{\beta}r(\mathbf{x}_0,y)\right)\right\|_2^2\right]$$

$$=\int_{\mathbf{x}_t}\int_y\left\{\int_{\mathbf{x}_0}p(\mathbf{x}_0|\mathbf{x}_t,y)\left\|h_{\boldsymbol{\psi}}\left(\mathbf{x}_t,y,t\right)-\exp\left(\frac{1}{\beta}r(\mathbf{x}_0,y)\right)\right\|_2^2 d\mathbf{x}_0\right\}p(\mathbf{x}_t|y)p(y)dyd\mathbf{x}_t$$

$$=\int_{\mathbf{x}_t}\int_y\left\{\|h_{\boldsymbol{\psi}}(\mathbf{x}_t,y,t)\|_2^2-2\langle h_{\boldsymbol{\psi}}(\mathbf{x}_t,y,t),\int_{\mathbf{x}_0}p(\mathbf{x}_0|\mathbf{x}_t,y)\exp\left(\frac{1}{\beta}r(\mathbf{x}_0,y)\right)d\mathbf{x}_0\rangle\right\}p(\mathbf{x}_t|y)p(y)dyd\mathbf{x}_t+C$$

$$=\int_{\mathbf{x}_t}\int_y\left\|h_{\boldsymbol{\psi}}(\mathbf{x}_t,y,t)-\mathbb{E}_{p(\mathbf{x}_0|\mathbf{x}_t,y)}\left[\exp\left(\frac{1}{\beta}r(\mathbf{x}_0,y)\right)\right]\right\|_2^2 p(\mathbf{x}_t|y)p(y)dyd\mathbf{x}_t,$$

where $C$ is a constant independent of $\boldsymbol{\psi}$. Thus we have the minimizer $\boldsymbol{\psi}^* = \arg\min_{\boldsymbol{\psi}} \mathcal{L}_{\text{guidance}}(\boldsymbol{\psi})$ satisfies

$$h_{\boldsymbol{\psi}^*}\left(\mathbf{x}_t,y,t\right)=\mathbb{E}_{p(\mathbf{x}_0|\mathbf{x}_t,y)}\left[\exp\left(\frac{1}{\beta}r(\mathbf{x}_0,y)\right)\right]. \qquad \square$$

## B More Details on Experiments

### B.1 Algorithms for Training the Guidance Network

Algorithm 1 is the algorithm for training the guidance network.

---

**Algorithm 1** Algorithm for Training a Guidance Network

---

**Require:** Samples from alignment dataset, pre-trained one-step diffusion model $s(\mathbf{x}_T, y, T)$, pre-determined reward function $r(\mathbf{x}_0, y)$, hyperparameters $\eta, \beta$, and initial weights of guidance network $\boldsymbol{\psi}$.

1: **repeat**
2:     Sample mini-batch data from alignment dataset with batch size $b$.
3:     Perturb $\mathbf{x}_0$ using forward transition $p(\mathbf{x}_T|\mathbf{x}_0)$.
4:     Compute guidance loss:
5:

$$\mathcal{L}_{\text{guidance}}(\boldsymbol{\psi}) = \frac{1}{b}\sum_{\mathbf{x}_0,\mathbf{x}_T,y}\left\|h_{\boldsymbol{\psi}}\left(\mathbf{x}_T,y\right)-\exp\left(\frac{1}{\beta}r(\mathbf{x}_0,y)\right)\right\|_2^2.$$

6:     Sample mini-batch from winning responses $(\mathbf{x}', y)$ with batch size $b$.
7:     Perturb $\mathbf{x}_0'$ using forward transition $q(\mathbf{x}_T'|\mathbf{x}_0')$.
8:     Compute consistency loss:
9:

$$\mathcal{L}_{\text{consistence}} = \frac{1}{b}\sum_{\mathbf{x}_0',\mathbf{x}_T',y}\left\|s(\mathbf{x}_T',y,T)+\nabla_{\mathbf{x}_T'}\log h_{\boldsymbol{\psi}}(\mathbf{x}_T',y)-\nabla_{\mathbf{x}_T'}\log q(\mathbf{x}_T|\mathbf{x}_0',y)\right\|_2^2.$$

10:     Update $\boldsymbol{\psi}$ via gradient descent:

$$\nabla_{\boldsymbol{\psi}}\left(\mathcal{L}_{\text{guidance}}+\eta\,\mathcal{L}_{\text{consistence}}\right).$$

11: **until** convergence
12: **return** weights of guidance network $\boldsymbol{\psi}$.

---

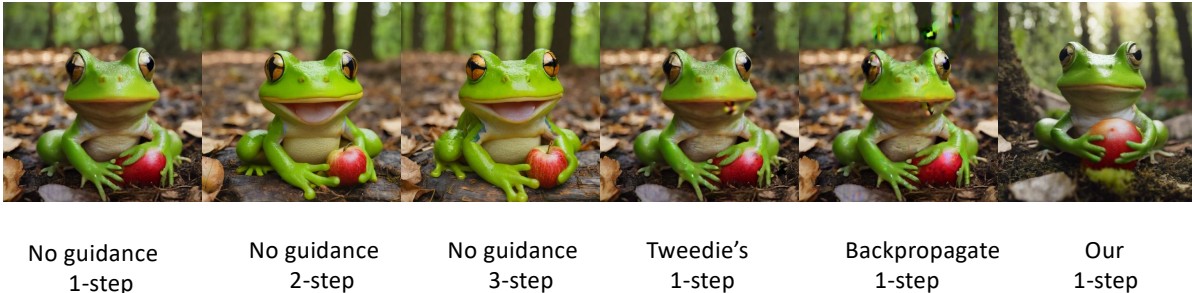

| No guidance 1-step | No guidance 2-step | No guidance 3-step | Tweedie's 1-step | Backpropagate 1-step | Our 1-step |

Figure 3: Effectiveness of the proposed method for diffusion models: The results demonstrate that 2-step and 3-step generation significantly improve the quality of the generated images compared to one-step generation. While two vanilla guidance methods (Tweedie's formula or directly backpropagation summarized in Section 3.2) fail to produce meaningful changes in the scene despite appropriate guidance strength, our method successfully achieves this enhancement. The prompt is "A photo of a frog holding an apple while smiling in the forest".

## B.2 Ablation Study on Hyperparameter

In this subsection, we provide the ablation study of the strength of the regularization $\eta$ and the strength of the reward function $\beta$ in the following table.

Table 4: Ablation study of hyperparameter on PickScore.

| $\eta$ | $\beta = 10$ | $\beta = 15$ | $\beta = 20$ |
|---|---|---|---|
| 0.1 | 22.82 | 22.79 | 22.72 |
| 0.5 | 22.78 | 23.01 | 22.79 |
| 1 | 22.76 | **23.08** | 22.84 |

## B.3 More Experimental Results

We provide the comparison between the proposed finetuning-free framework for diffusion models with two vanilla guidance methods in Fig 3.

## B.4 Prompts for Figure in Main Paper

Table 5: Prompts used to generate Figure 1.

| Image | Prompt |
|---|---|
| Col1 | Saturn rises on the horizon. |
| Col2 | a watercolor painting of a super cute kitten wearing a hat of flowers |
| Col3 | A galaxy-colored figurine floating over the sea at sunset, photorealistic. |
| Col4 | fireclaw machine mecha animal beast robot of horizon forbidden west horizon zero dawn bioluminiscence, behance hd by jesper ejsing, by rhads, makoto shinkai and lois van baarle, ilya kuvshinov, rossdraws global illumination |
| Col5 | A swirling, multicolored portal emerges from the depths of an ocean of coffee, with waves of the rich liquid gently rippling outward. The portal engulfs a coffee cup, which serves as a gateway to a fantastical dimension. The surrounding digital art landscape reflects the colors of the portal, creating an alluring scene of endless possibilities. |
| Col6 | A profile picture of an anime boy, half robot, brown hair |
| Col7 | Detailed Portrait of a cute woman vibrant pixie hair by Yanjun Cheng and Hsiao-Ron Cheng and Ilya Kuvshinov, medium close up, portrait photography, rim lighting, realistic eyes, photorealism pastel, illustration |
| Co18 | On the Mid-Autumn Festival, the bright full moon hangs in the night sky. A quaint pavilion is illuminated by dim lights, resembling a beautiful scenery in a painting. Camera type: close-up. Camera lens type: telephoto. Time of day: night. Style of lighting: bright. Film type: ancient style. HD. |

## B.5   Inference cost for Flow Matching Guidance

In practice, we find that using (S=3) endpoint samples already achieves strong performance, while larger values of (S) can further improve generation quality at the cost of increased computation. Moreover, the same set of endpoint samples can be reused throughout the entire denoising trajectory, so no additional endpoint sampling is required at each sampling step. As a result, the inference overhead remains modest. Compared with the vanilla model, our guidance incurs approximately $1.9\times$ the inference time, requiring 8.28 s/image versus 4.37 s/image on a single NVIDIA H200 GPU.

## B.6   None Differentiable Reward

Since the proposed framework is model agnostic and reward agnostic. Our method can be applied to any one-step model and even a non-differentiable reward function. We adopt the GenEval dataset to further demonstrate the effectiveness of the proposed method. The GenEval dataset evaluates whether the generated images are aligned with the prompt regarding object co-occurrence, position, count, and color. We apply official GenEval scripts to generate 5k training prompts. We use SDXL-turbo to generate 10 images per prompt to construct the source dataset and select the correct text image pair as the target dataset for regularization. We train the guidance network for 10 epochs and get the results in Table 6. It verifies the general applicability of the proposed framework. Most importantly, the reward function of the GenEval dataset is binary (1 for correct, 0 for incorrect), which is not differentiable. The unbiased Monte Carlo estimation of the direct backpropagation method cannot be applied to this non-differentiable reward function.

Table 6: Performance on the GenEval benchmark. Our method consistently outperforms SDXL across all sub-tasks.

| Method | Single Obj. | Two Obj. | Counting | Colors | Position | Color Attr. | Overall |
|--------|-------------|----------|----------|--------|----------|-------------|---------|
| SDXL | 0.97 | 0.72 | 0.37 | 0.83 | 0.10 | 0.21 | 0.53 |
| Ours | **0.98** | **0.75** | **0.41** | **0.86** | **0.16** | **0.26** | **0.57** |

