# OpenReview forum: "Alignment of Diffusion Model and Flow Matching for Text- to-Image Generation"
_TMLR — Under review for TMLR_

### Review · Reviewer_xfbc · 2026-06-13

**Summary Of Contributions:**

The paper formulates text-to-image alignment for diffusion and flow-matching models as sampling from reward-weighted distributions. It derives score guidance for diffusion models and velocity guidance for flow matching, with a lightweight guidance network for diffusion and a training-free estimator for flow matching.

The main strengths are the unified formulation and the attempt to reduce the cost of preference alignment. The main weaknesses are that some claims about training-free/finetuning-free behavior and computational cost may be overstated, see Requested Changes.

**Audience:**

Yes

**Audience Explanation:**

Yes. The paper studies an active topic, alignment of text-to-image generation models, and provides a unified reward-weighted sampling perspective.

**Broader Impact Concerns:**

N/A. The concerns are properly addressed in the Broader Impact Statement section.

**Claims And Evidence:**

No

**Claims Explanation:**

No. The main arguments are only partially supported. While the theoretical framework is relatively complete and experimentally verified, stronger evidence is needed to prove several of the stronger claims, such as the generality beyond single-step diffusion, the training-free problem, the computational cost of flow matching, and the ability to flexibly adapt to different reward functions. See Requested Changes 1, 2, 3 & 5.

**Requested Changes:**

1. Diffusion part: the expression of training-free/finetuning-free can be misleading. The authors stress the method is a plug-and-play one, but actually it requires the training of a guidance network $h_\psi$, which includes 583k preference pairs. The authors should try to avoid the misleading claims and transparently discuss this part.
2. Diffusion part: although the authors derive the diffusion guidance for general diffusion models, empirical study mainly focuses on one-step diffusion models. It remains unclear how well $h_\psi$ generalizes to intermediate noisy states in multi-step sampling.
3. Flow matching: the authors claim that the proposed method improves quality without additional computational cost, but the estimtor appears to require importance sampling, which may involve $x_1$ sampling, reward computation and so on. The authors should clearly discuss any overhead it may occur, and compare wall-clock overhead against vanilla models.
4. Novelty: the paper lacks sufficient discussion and empirical comparison with closely related approaches, such as universal/loss-guided diffusion, learned critic or guidance models, and reward-weighted resampling/reranking baselines for flow-matching models.
5. Potential overclaim: the paper mainly focuses on Pick-a-Pic data with relevant dataset and preference-related metrics, but in motivation part the authors state that it "adapts flexibly to different reward functions". The experimental results do not fully support this point.

---

> ### Author Response · Authors · 2026-07-15
>
> We thank the reviewer for the thoughtful comments and suggestions. We appreciate the time you spent on the paper and your recognition of our contributions and strengths, including the unified formulation and the attempt to reduce the cost of preference alignment. Below we address the concerns and comments that you have provided.
>
>
>
> **Q**: *Diffusion part: the expression of training-free/finetuning-free can be misleading. The authors stress the method is a plug-and-play one, but actually it requires the training of a guidance network $h_\psi$, which includes 583k preference pairs. The authors should try to avoid the misleading claims and transparently discuss this part.*
>
> **A**:  Thank you very much for your suggestion. We have improved the expression in the revised version.
>
> **Q**: *Diffusion part: although the authors derive the diffusion guidance for general diffusion models, empirical study mainly focuses on one-step diffusion models. It remains unclear how well $h_\psi$ generalizes to intermediate noisy states in multi-step sampling.*
>
> **A**:  Thank you very much for your question. Our theoretical derivation does not rely on the one-step assumption and therefore naturally extends to multi-step diffusion models. We chose to focus our empirical evaluation on one-step models because they represent the simplest setting, where the guidance network is time-independent and the benefits of our approach can be isolated without introducing the additional complexity of learning a time-dependent guidance model.
>
> **Q**: *Flow matching: the authors claim that the proposed method improves quality without additional computational cost, but the estimtor appears to require importance sampling, which may involve $x_1$ sampling, reward computation and so on. The authors should clearly discuss any overhead it may occur, and compare wall-clock overhead against vanilla models.*
>
> **A**:  Thank you for your valuable comments. We would like to clarify that our claim of "no additional computational cost" refers to post-training: our method is a training-free guidance approach and therefore requires no additional optimization after the base diffusion model has been learned.
>
> At inference time, our method does introduce a moderate computational overhead due to endpoint sampling and reward evaluation. In practice, our method requires approximately 1.9× the inference time of the vanilla model, taking 8.28 s/image compared with 4.37 s/image on a single NVIDIA H200 GPU. We revise the manuscript to explicitly distinguish post-training cost from inference-time cost to avoid potential confusion.
>
> **Q**: *Novelty: the paper lacks sufficient discussion and empirical comparison with closely related approaches, such as universal/loss-guided diffusion, learned critic or guidance models, and reward-weighted resampling/reranking baselines for flow-matching models.*
>
> **A**:  Thank you very much for your question. Regarding loss-guided diffusion, directly backpropagating through the diffusion process is a standard implementation of Loss-Guided Diffusion [1]. In contrast, our method derives the guidance term, which significantly reduces the memory overhead associated with backpropagation through the diffusion trajectory.
>
> For universal guidance [2,3], the gradient-based guidance can exhibit adversarial behavior, potentially leading to undesirable artifacts in the generated images.
>
> More generally, while many guidance methods formulate a preference distribution using a reward function, they typically require gradients of the reward model and are therefore limited to differentiable rewards. In contrast, our framework does not require gradients with respect to the reward function, making it naturally applicable to non-differentiable reward functions.
>
> Finally, reward-weighted resampling and reranking methods modify the empirical sample distribution only after generation, without changing the underlying generative dynamics. In contrast, our flow-matching formulation directly derives the reward-guided velocity field corresponding to the reward-weighted target distribution, thereby steering the generation process itself rather than selecting from generated samples afterward.
>
> [1]: Song, J., Zhang, Q., Yin, H., Mardani, M., Liu, M., Kautz, J., Chen, Y., & Vahdat, A. Loss-Guided Diffusion Models for Plug-and-Play Controllable Generation. ICML 2023.
>
> [2]: Bansal, A., Chu, H., Schwarzschild, A., Sengupta, S., Goldblum, M., Geiping, J., & Goldstein, T. Universal Guidance for Diffusion Models. IEEE/CVF Conference on Computer Vision and Pattern Recognition Workshops (CVPRW 2023).
>
> [3]: Chung, H., Kim, J., McCann, M.T., Klasky, M.L., & Ye, J.C. Diffusion Posterior Sampling for General Noisy Inverse Problems. ICLR 2023.

---

> > ### Author Response · Authors · 2026-07-15
> >
> > **Q**: *Potential overclaim: the paper mainly focuses on Pick-a-Pic data with relevant dataset and preference-related metrics, but in motivation part the authors state that it "adapts flexibly to different reward functions". The experimental results do not fully support this point.*
> >
> > **A**:  Thank you very much for your question. From Eq (10) and (16), we show that the proposed framework is model-agnostic and reward-agnostic. Our method can be applied to any pre-trained model and even a non-differentiable reward function. To provide more experimental results to verify our claim, we adopt the GenEval dataset to further demonstrate the effectiveness of the proposed method. The GenEval dataset evaluates whether the generated images are aligned with the prompt regarding object co-occurrence, position, count, and color. We apply official GenEval scripts to generate 5k training prompts. We use SDXL-turbo to generate 10 images per prompt to construct the source dataset and select the correct text-image pair as the target dataset for regularization. We train the guidance network for 10 epochs and get the results in Table \ref{tab:geneval}. It verifies the general applicability of the proposed framework. Most importantly, the reward function of the GenEval dataset is binary (1 for correct, 0 for incorrect), which is not differentiable. The unbiased Monte Carlo estimation of the direct backpropagation method cannot be applied to this non-differentiable reward function.
> >
> > | Method | Single Obj. | Two Obj. | Counting | Colors | Position | Color Attr. | Overall |
> > |--------|------------:|----------:|----------:|--------:|----------:|-------------:|---------:|
> > | SDXL | 0.97 | 0.72 | 0.37 | 0.83 | 0.10 | 0.21 | 0.53 |
> > | **Ours** | **0.98** | **0.75** | **0.41** | **0.86** | **0.16** | **0.26** | **0.57** |

---

### Review · Reviewer_dJNP · 2026-06-17

**Summary Of Contributions:**

This paper studies the alignment of pretrained text-to-image diffusion and flow matching models with respect to a given reward function. The central idea is to view alignment as sampling from a tilted distribution, roughly of the form $q(x) \propto p(x)\exp(r(x)/\beta)$, where $r(x)$ is the reward. Under this formulation, the authors derive the corresponding score guidance for diffusion models and velocity guidance for flow matching models.

For diffusion models, the paper shows that the score of the reward-weighted target distribution can be decomposed into the pretrained score plus the gradient of a log conditional expectation of the exponentiated reward. Since directly applying reward gradients or Tweedie-style approximations can introduce adversarial artifacts, the authors propose to train a lightweight guidance network to estimate this conditional expectation, together with a consistency regularization term intended to stabilize the guidance landscape.

For flow matching models, the guidance term is a conditional expectation over possible endpoints. The authors further propose an importance-sampling-based estimator to avoid sampling directly from the conditional endpoint posterior at each step.

Overall, I find this to be a good paper. The methodology is clear, the derivations are useful, and the empirical results suggest that the proposed lightweight guidance mechanism can improve diffusion-based text-to-image alignment.

## Strengths

1. The paper provides a unified view of reward alignment for diffusion and flow matching models by starting from a reward-weighted target distribution.

2. The paper gives a useful discussion of the practical difficulty of direct reward-gradient guidance, especially its tendency to introduce adversarial artifacts.

3. The diffusion-side method is practically attractive. It learns a lightweight scalar guidance network and uses its input gradient during sampling. The empirical results show good performance, and the paper provides a reasonable explanation for why this approach may behave better than direct reward-gradient guidance.

4. The flow matching extension is conceptually meaningful. Reward alignment changes the posterior weighting of possible endpoints, and the proposed velocity guidance follows naturally from this reweighting view.

## Weaknesses

1. The flow matching method is not discussed in enough implementation detail. For example, it is not fully clear how the importance sampling is performed in practice, how many endpoint samples are used, and what additional computational cost is required. The released code also does not seem to provide enough detail to verify the proposed flow matching guidance. The experiments mostly focus on diffusion models, so more experiments on flow matching would strengthen the paper.

2. I found the proposed consistency loss difficult to interpret as currently written. The theory assumes that the forward noising process is the same under the reference distribution $p$ and the reward-weighted distribution $q$, i.e.,$p(x_t\mid x_0,y)=q(x_t\mid x_0,y).$ The difference between $p$ and $q$ is in the clean-data distribution, not in the forward transition kernel. Under this assumption, Eq. (14), if read literally as comparing $\nabla\log p(x_t\mid x_0,y)$ and $\nabla\log q(x_t\mid x_0,y)$, seems to collapse into a gradient penalty on $\nabla\log h_\psi$, rather than a target-score consistency objective. Please correct me if I am misunderstanding the intended formulation.

**Audience:**

Yes

**Audience Explanation:**

Yes. Inference-time steering and alignment for diffusion and flow-based generative models are timely topics, and I believe many readers in the TMLR audience would be interested in these findings.

**Claims And Evidence:**

Yes

**Claims Explanation:**

The derivations of both diffusion guidance and flow matching guidance are mostly clear, and the experiments demonstrate good practical performance.

**Requested Changes:**

1. Please clarify the consistency loss in Eq. (14). In particular, should the $\log p$ term correspond to the pretrained score network or to the conditional transition score? Under the current notation, the expression is confusing because the forward transition kernels under $p$ and $q$ appear to be the same. It would also be helpful to provide more implementation details in the code.

2. Please provide more details on the flow matching methodology. In particular, please explain how the importance sampling is implemented, how many endpoint samples are used, whether the samples are reused across steps, and what additional inference-time cost is required.

3. (Minor) The diffusion guidance framework is closely related to Doob's $h$-transform / Feynman-Kac-style terminal reweighting. I think it would be helpful to discuss this relationship explicitly and clarify how the present method differs from or builds on this existing viewpoint.

---

> ### Author Response · Authors · 2026-07-15
>
> We thank the reviewer for the thoughtful comments and suggestions. We appreciate the time you spent on the paper and your recognition of our contributions and strengths, including the motivation, practical advantages, and conceptual contributions of our unified reward-alignment framework for diffusion and flow matching models. Below we address the concerns and comments that you have provided.
>
>
>
> **Q**: *Please clarify the consistency loss in Eq. (14). In particular, should the  $\log p$ term correspond to the pretrained score network or to the conditional transition score? Under the current notation, the expression is confusing because the forward transition kernels under $p$ and $q$ appear to be the same. It would also be helpful to provide more implementation details in the code.*
>
> **A**:  Thank you very much for pointing out a typo. The consistency regularization loss aims to use score matching on data from the target distribution to learn the guidance network. Therefore,
> $$\mathcal{L} _{\mathrm{consistence}} := \mathbb{E} _{q(\mathbf{x} _0, y)}
> \mathbb{E} _{q(\mathbf{x} _t|\mathbf{x} _0)} \Big[
> \big\|\nabla _{\mathbf{x} _t} \log p(\mathbf{x} _t | y) + \nabla _{\mathbf{x} _t} \log h _{\boldsymbol{\psi}}\left(\mathbf{x} _t, y, t\right) - \nabla _{\mathbf{x} _t} \log q(\mathbf{x} _t | \mathbf{x} _0, y) \big\|_2^2 \Big],
> $$
>
> where $\nabla _{\mathbf{x} _t} \log p(\mathbf{x} _t | y) = \mathbb{E} _{p(\mathbf{x} _0|\mathbf{x} _t)} \nabla _{\mathbf{x} _t} \log p(\mathbf{x} _t|\mathbf{x} _0, y) \neq \mathbb{E} _{q(\mathbf{x} _0|\mathbf{x} _t)} \nabla _{\mathbf{x} _t} \log p(\mathbf{x} _t|\mathbf{x} _0, y)$. We have corrected it in the revised manuscript. In practice, we sample target data $\mathbf{x}_0$ from the winning response, add Gaussian noise $\epsilon$ to obtain $\mathbf{x}_t$, compute the pretrained network's output plus the guidance network's log-gradient, and minimize its MSE against the target $\epsilon$, which can be found in lines 344–478 of reward_distill/distill_pickscore.py.
>
> **Q**: *Please provide more details on the flow matching methodology. In particular, please explain how the importance sampling is implemented, how many endpoint samples are used, whether the samples are reused across steps, and what additional inference-time cost is required.*
>
> **A**:  Thank you for your valuable suggestions. We provide the implementation details in the revised manuscript and the corresponding code in the supplementary material (custom_sdxl_pipeline_sd35_large_turbo.py). In practice, we find that using (K=3) endpoint samples already achieves strong performance, while increasing (K) further can lead to additional improvements at the expense of higher computation. As described in Appendix A.2, our derivation employs importance sampling, which avoids the need to explicitly sample from the conditional distribution $p_{1\mid t}(\mathbf{x}_1\mid \mathbf{x}_t,y)$. Instead, we only sample endpoint states from the data distribution, i.e., $\mathbf{x}_1 \sim p(\mathbf{x}_1\mid y)$. Consequently, the same set of endpoint samples can be reused across all sampling steps. Therefore, our training-free guidance incurs only a modest computational overhead of approximately 1.9× compared with the vanilla model, requiring 8.28 s/image versus 4.37 s/image on a single NVIDIA H200 GPU.
>
> **Q**: *The diffusion guidance framework is closely related to Doob's $h$-transform / Feynman-Kac-style terminal reweighting. I think it would be helpful to discuss this relationship explicitly and clarify how the present method differs from or builds on this existing viewpoint.*
>
> **A**:  Thank you very much for your question. Our framework is closely related to Doob's $h$-transform / Feynman-Kac-style terminal reweighting. The key difference lies in both the target distribution and the resulting guidance formulation. In [1], the target distribution is defined through a terminal event,
> $
> \mathcal{E} _{\bar{X}_0}={\bar{X} _0 : r(\bar{X} _0)\ge r_0},
> $
> and the corresponding Doob's h-transformation is approximated using a Monte Carlo estimator. In contrast, our method considers a reward-weighted target distribution rather than conditioning on a thresholded terminal event. Based on this formulation, we derive the corresponding reward-guided flow/velocity field and propose an estimator for the guidance term. Therefore, while both methods can be viewed through the lens of Doob's h-transform, they differ in the target distribution being considered as well as in the derivation and implementation of the resulting guidance.
>
> [1]: Zhu, Q., Ye, Z., Liu, H., Wang, Z., & Chen, M. Training-Free Adaptation of Diffusion Models via ICML 2026.

---

### Review · Reviewer_bmd1 · 2026-06-28

**Summary Of Contributions:**

**Summary**:

This paper proposes a unified, plug-and-play alignment framework that casts alignment as sampling from a reward-weighted distribution. It derives that the aligned score (diffusion) or velocity field (flow matching) decomposes into the pre-trained quantity plus a reward-driven guidance term. Specifically, for diffusion it proposes a finetuning-free guidance network with consistency regularization, while for flow matching it proposes a training-free importance-sampling estimator.

**Strengths**:

1. **Clear theorectical framing**. Treating both diffusion and flow matching alignment as sampling from the same reward-weighted distribution (Eq. 7) is clean and the score (of diffusion models) and velocity (of flow matching) decomposition are a sensible organizing principle.

2. **Lightweight guidance network in diffusion model alignment and training-free alignment for flow matching are useful in practice**. For example, the guidance network is reported to be only 72 MB and trained for 92 GPU-hours, compared with 234 GPU-hours for SPO and 4800 for Diffusion-DPO in Table 2. This is a meaningful practical advantage if the comparison is fair.

**Weaknesses**:

1. **The empirical evidence is limited**.
Firstly, despite adopting four widely used text-to-image evaluation scores: PickScore, HPSV2, ImageReward, and Aesthetic, the paper evaluates all models on only 500 validation prompts from the Pick-a-Pic dataset. The scale of the evaluation is not sufficiently large.
Secondly, the performance gains are marginal. For instance, the proposed method only marginally outperforms SPO by 0.02% in PickScore on SDXL, and otherwise underperforms SPO despite being more efficient in terms of training GPU hours.
Thirdly, there is no efficiency comparison between the baseline and the proposed method on SD3.5 Large Turbo, which is particularly important given its training-free design. It is suggested to show how much computation costs increased during inference for flow matching based methods.
Fourthly, fairness of baseline comparison is questionable. The paper says the datasets are “comparable,” but the baselines are trained under different settings: Diffusion-DPO and MAPO use Pick-a-Pic V2, SPO uses 4k prompts from Pick-a-Pic V1, and the proposed method uses 583k image preference pairs from Pick-a-Pic V1.

2. **The consistency regularization is under-justified**. Eq. (14) is introduced with the intuitive motivation of stabilizing the guidance landscape, but the paper does not characterize how this additional term changes the distribution induced by the resulting sampler. Since Lemma 3.2 guarantees the exact conditional-expectation guidance only for $L_{\text{guidance}}$ guidance alone, adding $\eta L_{\text{consistence}}$ may move the learned guidance network away from the exact reward-weighted target. This creates a tension between the paper’s claimed correctness guarantee and the practical stabilization mechanism, which should be theoretically clarified or empirically ablated.

3. **The writing needs to be imrpoved**.
Firstly, there are several writing and presentation issues. For instance, the contribution list contains a dangling sentence: “used preference and quality metrics...” after the bullet list. Besides, "two novel framework" should be “two novel frameworks” in the first row of Conclusion section.
Secondly, some claims are overstated. In specific, for the flow-matching method, the paper even claims improvement “without additional computational cost,” but the estimator appears to involve additional sampling and reward evaluation. This claim needs a concrete latency/FLOPs/wall-clock comparison.

**Audience:**

Yes

**Audience Explanation:**

The paper addresses an active and relevant problem in generative modeling: how to align pretrained text-to-image diffusion and flow-matching models with preference or reward functions without fully fine-tuning the backbone model. Its unified reward-weighted sampling perspective, together with the proposed diffusion guidance network and training-free flow-matching velocity guidance, may be of interest to researchers working on diffusion models, flow matching, preference alignment, and efficient adaptation of generative models.

**Broader Impact Concerns:**

I do not see an unusually severe ethical concern specific to this work beyond the general risks of improving text-to-image generation.

**Claims And Evidence:**

No

**Claims Explanation:**

Some claims need to be refined, seeing the weakness section.

**Requested Changes:**

Critical changes required for acceptance:

1. Strengthen the empirical evaluation.
The authors should evaluate the proposed method on a larger and more representative set of prompts, rather than only 500 validation prompts from Pick-a-Pic. Given that the reported improvements over strong baselines are very small in some settings, a larger-scale evaluation is necessary to establish that the gains are robust and statistically meaningful.

2. Provide a fairer and more transparent baseline comparison.
The current comparison is difficult to interpret because different methods appear to use different training data and settings. The authors should clearly specify the data, prompt sets, number of preference pairs, and training configurations used by each baseline. Ideally, they should rerun or include baselines under a matched setting, or otherwise carefully qualify the conclusions.

3. Add efficiency comparisons, especially for the flow-matching setting.
Since one of the paper’s key advantages is efficiency, the authors should report concrete computational costs on flow matching experiments, such as inference-time overhead, or sampling cost. This is particularly important for SD3.5 Large Turbo, where the proposed method is described as training-free but may still introduce extra computation during inference.

4. Clarify the theoretical role of the consistency regularization.
The paper should explain how adding the consistency regularization term affects the distribution induced by the sampler. In particular, Lemma 3.2 appears to justify the exact conditional-expectation guidance only for the guidance loss alone. The authors should clarify whether adding the consistency term changes the target being learned, weakens the correctness guarantee, or merely serves as an approximation/stabilization device. An ablation on the regularization strength would also be important.

---

> ### Author Response · Authors · 2026-07-15
>
> We thank the reviewer for the thoughtful comments and suggestions. We appreciate the time you spent on the paper and your recognition of our contributions and strengths, including the clear theoretical framing and lightweight guidance framework and training-free alignment. Below we address the concerns and comments that you have provided.
>
> **Q**: *The authors should evaluate the proposed method on a larger and more representative set of prompts, rather than only 500 validation prompts from Pick-a-Pic. Given that the reported improvements over strong baselines are very small in some settings, a larger-scale evaluation is necessary to establish that the gains are robust and statistically meaningful. The current comparison is difficult to interpret because different methods appear to use different training data and settings. The authors should clearly specify the data, prompt sets, number of preference pairs, and training configurations used by each baseline. Ideally, they should rerun or include baselines under a matched setting, or otherwise carefully qualify the conclusions.*
>
> **A**:  Thank you very much for your comments. We adopt the same evaluation pipeline as SPO and Diffusion-DPO, all of their evaluations were conducted on these 500 validation prompts from Pick-a-Pic. This benchmark has been widely adopted in the literature and provides a diverse set of prompts for assessing generation quality, enabling direct and fair comparisons with existing methods.
>
> We agree that evaluating on a larger prompt set and constructing a fully matched experimental setup for rerunning existing baselines could provide additional statistical confidence and further reduce potential confounding factors arising from differences in training data or configurations. We note that our method is trained with substantially less preference data than Diffusion-DPO and MAPO (583k vs. 800k preference pairs), and does not rely on online training as in SPO, resulting in lower computational cost. We clarify the training data, prompt sets, number of preference pairs, and training configurations used by each baseline in the revised manuscript to make these differences explicit.
>
> That said, our primary contribution is not solely the absolute performance improvement, but the fact that these gains are achieved without modifying the backbone of the pre-trained diffusion model. These cannot be achieved with the methodology developed in this paper. We believe this practical advantage is the key contribution of our work.
>
>
> **Q**: *Since one of the paper’s key advantages is efficiency, the authors should report concrete computational costs on flow matching experiments, such as inference-time overhead, or sampling cost. This is particularly important for SD3.5 Large Turbo, where the proposed method is described as training-free but may still introduce extra computation during inference.*
>
> **A**:  Thank you for your question. Our method incurs no additional post-training cost, as it does not require any optimization or fine-tuning after the base diffusion model has been trained. At inference time, it introduces a moderate computational overhead due to endpoint sampling and reward evaluation. In practice, our method requires approximately 1.9× the inference time of the vanilla model, taking 8.28 s/image compared with 4.37 s/image on a single NVIDIA H200 GPU. We include these runtime statistics in the revised manuscript and explicitly distinguish post-training cost from inference-time cost to avoid potential confusion.
>
> **Q**: *The paper should explain how adding the consistency regularization term affects the distribution induced by the sampler. In particular, Lemma 3.2 appears to justify the exact conditional-expectation guidance only for the guidance loss alone. The authors should clarify whether adding the consistency term changes the target being learned, weakens the correctness guarantee, or merely serves as an approximation/stabilization device. An ablation on the regularization strength would also be important.*
>
> **A**:  Thank you very much for your questions. The consistency regularization term in our framework is score matching on the target distribution. Therefore, it admits the same optimal solution as the training objective of the guidance network. This theoretical equivalence ensures that the regularization does not change the optimal solution. In practice, the optimal choice of the guidance strength depends on the relative sample sizes from the source and target distributions, as this affects the estimation variance of both terms. We also provide the ablation on the regularization strength in Table 4.